# Open Weather and Climate Science in the Digital Era

Martine G. de Vos[1,2], Wilco Hazeleger[1,3], Driss Bari[4], Joerg Behrens[5], Sofiane Bendoukha[5], Irene Garcia-Marti[6], Ronald van Haren[1], Sue Ellen Haupt[7], Rolf Hut[8], Fredrik Jansson[9], Andreas Mueller[10], Peter Neilley[11], Gijs van den Oord[1], Inti Pelupessy[1], Paolo Ruti[12], Martin G. Schultz[13], and Jeremy Walton[14]

[1]Netherlands eScience center, Amsterdam, the Netherlands
[2]Information and Technology Services, Utrecht University, Utrecht, the Netherlands
[3]Geosciences, Utrecht University, Utrecht, the Netherlands
[4]CNRMSI/SMN, Direction de la Meteorologie Nationale Casablanca, Morocco
[5]German Climate Computing Centre (DKRZ), Hamburg, Germany
[6]Royal Netherlands Meteorological Institute (KNMI), De Bilt, the Netherlands
[7]Research Applications Laboratory, National Center for Atmopsheric Research, Boulder, USA
[8]Water Resources Management, Delft University of Technology, Delft, the Netherlands
[9]Centrum Wiskunde & Informatica, Amsterdam, the Netherlands
[10]Numerical methods, European Centre for Medium-Range Weather Forecasts, Reading, UK
[11]The Weather Company/IBM, Boston MA, USA
[12]World Weather Research Division, World Meteorological Organization, Geneva, Switzerland
[13]Jülich Supercomputing Centre, Forschungszentrum Jülich, Jülich, Germany
[14]Hadley Centre for Climate Science, Met Office, Exeter, UK

**Correspondence:** Martine G. de Vos (m.g.devos@uu.nl)

**Abstract.** The need for open science has been recognized by the communities of meteorology and climate science. While these domains are mature in terms of applying digital technologies, the implementation of open science methodologies is less advanced. In a session on "Weather and Climate Science in the Digital Era" at the 14th IEEE International eScience Conference domain specialists and data and computer scientists discussed the road towards open weather and climate science.

Roughly 80% of the studies presented in the conference session showed the added value of open data and software. These studies included open datasets from disparate sources in their analyses, or developed tools and approaches that were made openly available to the research community. Furthermore, shared software is a prerequisite for the studies which presented systems like a model coupling framework or digital collaboration platform. Although these studies showed that sharing code and data is important, the consensus among the participants was that this is not sufficient to achieve open weather and climate

science and that there are important issues to address.

At the level of technology, the application of the FAIR principles to many datasets used in weather and climate science remains a challenge. This may be due to scalability (in the case of high-resolution climate model data, for example), legal barriers such as those encountered in using weather forecast data, or issues with heterogeneity (for example, when trying to make use of citizen data). In addition, the complexity of current software platforms often limits collaboration between

researchers and the optimal use of open science tools and methods.

The main challenges we observed, however, were non-technical and impact the practice of science as a whole. There is a need for new roles and responsibilities in the scientific process. People working at the interface of science and digital technology - e.g., data stewards and research software engineers - should collaborate with domain researchers to ensure the optimal use of open science tools and methods. In order to remove legal boundaries on sharing data, non-academic parties such as meteorological institutes should be allowed to act as trusted agents. Besides the creation of these new roles, novel policies regarding open weather and climate science should be developed in an inclusive way in order to engage all stakeholders.

Although there is an ongoing debate on open science in the community, the individual aspects are usually discussed in isolation. Our approach in this paper takes the discourse further by focusing on 'open science in weather and climate research' as a whole. We consider all aspects of open science and discuss the challenges and opportunities of recent open science developments in data, software and hardware. We have compiled these into a list of concrete recommendations that could bring us closer to open weather and climate science. We acknowledge that the development of open weather and climate science requires effort to change, but the benefits are large. We have observed these benefits directly in the studies presented in the conference and believe that it leads to much faster progress in understanding our complex world.

## 1  INTRODUCTION

In this article we describe the main findings of a conference session on "Weather and Climate Science in the Digital Era" with a special focus on the implementation of open science methodologies.

Meteorology and climate sciences are data- and computationally-intensive areas of research by tradition. Being primarily a physical science, empirical data collection has always been important and meteorology was one of the first fields that standardized data collection from the advent of systematic instrumental observations in the mid-1800s (e.g. Maury, 1853; Quetelet, 1874). In addition, the production of meteorological forecasts was one of the first applications to be developed for electronic computers, following decades during which the calculations were performed by hand (we recall that "computer" originally meant "one who computes", and that the adjective "electronic" was introduced to distinguish the machine from the human). Numerical weather prediction (NWP) has advanced from the first operational predictions in the 1950s (Charney et al., 1950), aided by increased computing capability and the growing supply of observational data to generate initial conditions for assimilation into the model state. Climate research has benefitted from the same developments (see e.g. Lynch, 2008, for an overview).The assimilation of observational data into NWP models has been a turning point for the development of high-resolution gridded information of the atmosphere and ocean state (e.g. Kalnay et al., 1996; Dee et al., 2011). The use of this methodology for reanalysis - that is, generating a comprehensive and physically consistent record of how the weather is changing over time - has ensured a baseline for climate research and triggered the development of downstream climate services.

Meteorologists have been using machine learning to post-process model output, blend multiple models, and optimize the weighting of models for over 20 years (Haupt et al., 2018). Neural nets were used in the 90s to speed up the calculation of outgoing longwave radiation in climate models (Chevallier et al., 1999), and for both short- and long-wave radiation parameterization in the National Center for Atmospheric Research (NCAR) Community Atmospheric Model (CAM) (Krasnopolsky

et al., 2007). Present and future strategies feature an Earth System approach for assimilating environmental data into a more comprehensive coupled system including the atmosphere, ocean, biosphere and sea-ice (Penny and Hamill, 2017).

The influence and application of digital technologies has shown no sign of abatement in recent times. Three technological developments are having a strong effect on meteorology and climate research (Ruti et al., 2019). First, the increase of computing power. Exascale (i.e., $10^{18}$ operations per second) is the next proxy in the long trajectory of exponential performance increases that has continued for more than half a century (Reed and Dongarra, 2015) and provides unprecedented opportunities with regard to the finer resolution of scales in time and space, and/or the coupling of more components that represent different parts of the Earth system. However, it also poses large software development and data management challenges, such as the impact of increasing numerical model resolution, increasing code complexity, and the volumes of data that are handled (Bauer et al., 2015; Sellar et al., 2020). A second development concerns the open availability of standard meteorological data and data from a variety of sources, including citizen science projects and low-cost sensors. Modern data management tools enable handling these data sources. Thirdly, there has been increasing use of machine learning, in particular so-called deep learning. A plethora of machine learning methods have been and are being applied to problems of weather and climate prediction, from emulating unresolved processes in numerical models to calibrating forecasts produced with numerical models and the production of forecasts based on data and machine learning methods only (Huntingford et al., 2019; Schneider et al., 2017; Reichstein et al., 2019).

Digital technologies enable new research methods, accelerate the growth of knowledge, and spur the creation of new means of communicating such knowledge amongst researchers and within the broader scientific community. As such, these technologies have reshaped the scientific enterprise and are strongly connected to open science (OECD, 2015; Bourne et al., 2012). Open science methodologies such as open access publications, open source software development and FAIR data (see below) stimulate the use of data and software resources and lead to more reproducible research (Wilkinson et al, 2016; Munafò et al., 2017). The need for open research practices has been recognized by the communities of meteorology and climate science. Nonetheless, whilst these domains are mature in terms of the application of digital technologies, the implementation of open science methodologies is less advanced.

In a session on "Weather and Climate Science in the Digital Era" at the 14th IEEE International eScience Conference, domain specialists and data and computer scientists discussed the road towards open weather and climate science. This paper describes the main findings and insights from this conference session.

The remainder of this paper is organized as follows: In the Methods section we describe the set-up of the conference session in detail, since the insights and claims in this paper are based on the observations made during the session. The Open Science section contains a small literature review which describes the progress of open weather and climate science in the context of open science developments in general. In the section Towards Open Weather and Climate Science we discuss the challenges and opportunities of open data and open software. The last section provides a synthesis of the issues that should be addressed in order to achieve open weather and climate science.

## 2 METHODS

The "Weather and Climate Science in the Digital Era" conference session examined some of the data and compute intensive approaches which are used in weather and climate science. The session comprised ten oral abstract presentations, one keynote talk, and six short poster pitches. Contributions were selected after a peer review on their scientific merit and innovative nature and published in the conference proceedings (Bari, 2018; Behrens et al., 2018; Bendoukha, 2018; Brangbour et al., 2018; Garcia-Marti et al., 2018; Haupt et al., 2018; Hut et al., 2018; Jansson et al., 2018; Pelupessy et al., 2018; Ramamurthy, 2018; Schultz et al., 2018; Stringer et al., 2018; van Haren et al., 2018; van den Oord et al., 2018). The sixteen session participants were either presenters or involved in the organization of the session, and represented disparate science domains, as well as computer and data sciences.

Following the first part of the session which was dedicated to the presentations, the participants broke into three groups to discuss "challenges and opportunities regarding open weather and climate science". The findings of each group were presented and discussed in a final plenary session, during which observations and insights were documented.

The observations in this paper are based on both the insights from the studies presented in the session, and the notes made during the discussion. The majority of the participants in the session also contributed to this paper. As such, this represents a shared view of a group of experts in weather and climate science on digital and open science developments in their field.

## 3 OPEN SCIENCE

Based on a small literature review, this section describes the progress of open weather and climate science in the context of open science developments in general.

Open science refers to open research practices, and includes but is not limited to public access to the academic literature, sharing of data and code (Mckiernan et al., 2016). However, the interpretation of the concept of open science varies between different schools of thought (Fecher and Friesike, 2014). In general, open science concerns various stakeholders: besides scholars, these include institutes, research funders, librarians and archivists, publishers and decision makers (Bourne et al., 2012; OECD, 2015; Fecher and Friesike, 2014).

It has been shown that the adoption of open research practices leads to significant benefits for researchers: specifically, increases in citations, media attention, potential collaborators, job opportunities and funding opportunities (Mckiernan et al., 2016). Europe and the United States have made efforts to adapt legal frameworks and implement policy initiatives for greater openness in scientific research (OECD, 2015; National Science Foundation, 2018). Several countries provide digital infrastructure based on rich metadata that support the optimal re-use of resources in the research environment (Mons et al., 2017). Examples include the European Open Science Cloud in Europe (Directorate-General for Research and Innovation, 2018), NIH Data Commons projects in the United States, AARnet in Australia (AARNet, 2018) and the African Data Intensive Research Cloud in South Africa (R. Simmonds et al., 2016). Funders and research institutes have announced policies encouraging, mandating, or specifically financing open research practices (Mckiernan et al., 2016; Wilkinson et al, 2016) - for example, the National Science Foundation in the United States (National Science Board, 2011), CERN in Switzerland (CERN-OPEN-2014-

049, 2014), the Netherlands Organization for Scientific Research (executive board, 2019) and the United Nations Educational, Scientific and Cultural Organization (board, 2013).

The need for open research practices has been recognized by the communities of meteorology and climate science and has even entered into the political arena. For instance, in its report on the so-called "Climatic Research Unit email controversy" in 2009 the Science and Technology Committee of the UK House of Commons stated that climate science is a matter of
great importance and that the quality of the science should be irreproachable. The committee called for the climate science community to become more transparent by publishing raw data and detailed methodologies (House of Commons, 2010).

There are many examples of open access, open data and open source software in meteorology and climate science. The United States has a long history of making meteorological observations, model source codes and model output an open public commodity, available to all. The WRF regional model, MPAS global model, and the CESM climate model (Skamarock
et al., 2019; Hurrell, J.W. et al., 2013) are good examples of shared numerical weather and climate model codes. Output from NOAA weather and climate prediction models are freely available. The European Center for Medium-range Weather Forecasts (ECMWF) provides researchers with a free, and easy-to-use version of the Integrated Forecasting System (IFS), which is one of the main global NWP systems (Carver, 2019). It allows IFS to be used by a much wider community and the academic community contributes to improving the forecast model with new developments. The UK Earth System model (Sellar
et al., 2019), a joint development between the National Environment Research Council (NERC) and the UK Met Office, has been made available to the research community in a similar fashion.In addition, co-ordinated coupled model intercomparison projects (CMIP) (Taylor et al., 2012; Eyring et al., 2016) are excellent examples of the climate modeling community working together. The construction of multi-model comparisons and statistics forces research groups to accept common input forcings, provide detailed documentation of the numerical schemes in their model and produce open, standardized output data (Sellar
et al., 2020, see e.g.). The result is a better understanding of climate change arising from natural, unforced variability or in response to changes in radiative forcing in a multi-model context.

The international meteorological and climate research communities have been sharing data since the 1990s, using common file and metadata formats. Besides CMIP (Taylor et al., 2012), examples include the sharing of reanalysis data, starting with NCEP/NCAR reanalysis and ECMWFs ERA reanalysis data products (Dee et al., 2011; Kalnay et al., 1996, e.g.).
There is an ongoing debate on open science in the meteorology and climate research community, but in literature the individual open science practices are discussed separately. Elements have been discussed in literature, e.g. in Ruti et al. 2019 on strategic programming level, in Eyring et al 2020 2020 on a generic software tool for Earth system model data diagnostics, the open software platform PANGEO (Pangeo), and community simulation model as the regional model WRF and CESM (Skamarock et al., 2019; Hurrell, J.W. et al., 2013). Additionally, these aspects are discussed in Climate Informatics work-
shops (Cli), workshops held as part of the European Network on Earth System Modelling (ENES)(Modelling), workshops of operational centres as the European Centre for Medium-Range Weather Forecasting (e.g. the bi annual High Performance Computing workshop) to name a few.

The examples described above show that open research practices are growing in popularity and necessity. However, widespread adoption of these practices has not yet been achieved, which is also true for meteorology and climate science. In fact, sharing

of data, software and vocabularies is only common practice in a few fields such as astronomy and genomics (Consortium, 2004; Borgman, 2012; Shamir et al., 2013, e.g.). Recent studies show that transparency and reproducibility are still a matter of concern to the scientific community as a whole. It requires that all stakeholders work together to create a more open and robust system (Baker, 2016; Munafò et al., 2017; Gil et al., 2016).

## 4   TOWARDS *OPEN WEATHER AND CLIMATE SCIENCE*

In the following section we present our perspective on the challenges and opportunities regarding open weather and climate science.

### 4.1   OPEN DATA

About 50% of the studies reported in the proceedings of the conference session include open data from different sources in their analyses. Examples include the use of open satellite data, geolocated data via OpenStreetMap and openly available in-situ
meteorological observations (Haupt et al., 2018; Garcia-Marti et al., 2018; Bari, 2018; Schultz et al., 2018, and references therein). Two studies include data that are not common in meteorological or climate research. Citizen data such as social media posts (Brangbour et al., 2018) and observations from amateur weather stations (van Haren et al., 2018) can lead to new perspectives on local conditions beyond data from traditional meteorological stations.

At least 50% of the studies use common file formats and standard protocols to facilitate the exchange and use of data. Van den
Oord et al. 2018 use CF-netCDF formats. The CF conventions provide guidelines for the use of metadata in the netCDF file and are increasingly used in climate studies. Behrens et al. 2018, Pelupessy et al. 2018, Schultz et al. 2018 and Stringer et al. (2018) all use standard protocols for inter-process communication (like MPI and REST) in their numerical codes. Furthermore, the use of common file formats and standard protocols is a prerequisite for the digital collaboration platforms which were presented in the session (Ramamurthy, 2018; Hut et al., 2018; Bendoukha, 2018).

The session participants recognized that in the current weather and climate science community the focus is primarily on making data and software findable and accessible, often via web portals. Although these are necessary first steps towards open science, we acknowledge that these steps are not sufficient. Data and software that are findable and accessible may still be hard to obtain in practice or may be disseminated in a way that it is still difficult to interpret and use. Wilkinson and colleagues (2016) defined guidelines to ensure the transparency, reproducibility, and reusability of scientific data. These state that data
- and also the algorithms, tools, and workflows that led to these data - should be Findable, Accessible, Interoperable and Reusable (FAIR). The FAIR guidelines put specific emphasis on enhancing the ability of machines to automatically find and use the data, in addition to supporting its reuse by individuals.

In order to make the output from weather and climate models open and interoperable, i.e. formatted according to standards such as CF-netCDF,, including all necessary metadata, we consider performance scalability as the foremost technological
challenge. Whereas the simulation models are predominantly run on large clusters using many compute nodes, subsequent processing and analysis of the output is often still confined to a single CPU and does not scale easily with (say) increased

model resolution. Thus, producing FAIR model output via traditional post-processing pipelines is quickly becoming infeasible for advanced simulation models due to the sheer volume and complexity of their output.

For simulation models, this trend is a consequence of the advance of processor speed and model scalability compared to storage bandwidth, and can be countered with two strategies. The first is removing the need for post-processing by incorporating as many steps as possible within the application itself. This will make the model more expensive, especially in terms of memory usage, but the overhead may often be mitigated by offloading the post-processing to a small extra set of dedicated high-memory compute nodes. This approach requires a technical effort from the data providers in the community, and it can only solve the data problem to a limited extent, since there will always be extra manipulations required for many scientific analyses. Hence we need a second strategy on the data users' side to increase parallelism in the climate data processing toolchain. Existing cloud computing technologies, like Apache SPARK (Zaharia et al., 2016) or Dask (Team, 2016), may provide a suitable basis, since data processing and analysis pipelines can usually be represented by task graphs with a large degree of parallelism (over grid points, over multiple variables, over ensemble members, etc.). One of the key aspects, however, is the capability of the developer, usually a meteorologist or climate scientist, to adopt a new programming paradigm which facilitates the parallel execution of the workflow on cloud infrastructure. Here, research software engineers may play a key role by - for instance - developing higher-complexity algorithms for efficient processing of distributed climate data and adopting tools like xarray (Hoyer and Hamman, 2017) and Iris (Office, 2010).

In addition to these technological issues, we observe that some important challenges for open data arise from the political or legal context, and as such require additional efforts beyond the scientific domain. Weather services and commercial entities can see their data as a business advantage and be reluctant to make these open. Various resolutions by the World Meteorological Organisation (e.g. Resolution 40, 25 and 60) promote open access and exchange of data in order to better manage the risks from weather and climate-related hazards, but leave room for additional conditions. These resolutions have no legal status and national legislation may lead to restricted access to data and charges (Sylla, 2018). Also, policies to promote open data are less mature than those to promote open access to scientific publications (OECD, 2015). Another way to solve these issues is by signing nondisclosure agreements and allow the weather services to act as trusted agents who use the data for the public good without disclosing their details. These trusted agents should be considered as occupying a new role in the scientific process.

Furthermore, data need to be hosted and maintained, and their quality should be ensured. These requirements are well-addressed for large operational data services, such as the European Copernicus program, but this is not usually the case for research data of individual scientists, despite the increasing attention being paid to data management. Currently, data providers have no clear policy (such as - for example - the FAIR principles) to follow in their hosting and management of data. Publications such as Geoscience Data Journal, Scientific Data and Earth System Data, are a partial remedy as these provide open access platforms where scientific data can be peer-reviewed and formally published. Some funding agencies - for example NWO in the Netherlands - are now requiring that, for all projects they fund, software becomes open source and the data are archived and findable unless there are strong reasons not to do so (e.g. privacy). Also, research funded by the European Commission should adhere to FAIR principles and data management plans need to be in place.

## 4.2  OPEN SOFTWARE

The conference session provided excellent examples of tools and approaches that were developed and made openly available to the research community. For example, approaches to reduce the computational or post processing costs of existing simulation models (Stringer et al., 2018; Behrens et al., 2018; van den Oord et al., 2018; Jansson et al., 2018) and approaches to integrate data sets from different sources (van Haren et al., 2018; Schultz et al., 2018). Four studies in the session presented an approach for which open data and software is a prerequisite, as these comprise a model coupling framework or a digital collaboration platform (Pelupessy et al., 2018; Jansson et al., 2018; Ramamurthy, 2018; Hut et al., 2018; Bendoukha, 2018).

We strongly support open publication of code, even if this code is under development, and especially when this code is used in a paper to support research findings. Open code can be inspected and reused by peers; this improves the reproducibility and quality of the corresponding research. Code sharing is crucial to science and to climate research in particular, since local and global policies depend on the scientific results. Open publication, however, requires the code to be documented and tested, which is a time-consuming effort. This level of documentation and testing is not yet standard practice, partially because there is no incentive to do so. There is a need for open science practices where incentives are developed to share scientific information beyond the final result in a scientific paper. Agile (Fowler and Highsmith, 2001) is a well-known approach in the software engineering community, and may provide a means to achieve open scientific software in a feasible way. According to the Agile approach, software is developed in small increments every few weeks, which makes it possible to provide continuous feedback to the developers. With its focus on flexibility and communication, Agile lends itself naturally to scientific software projects which are characterized by frequent code alterations due to changing requirements, tight collaboration in small teams, and short planning horizons (Sletholt et al., 2012). Agile practices are used, for example, by the ECMWF to develop the Climate Data Store (Raoult et al., 2017) and the Met Office Hadley Centre to develop climate models (Easterbrook and Johns, 2009).

In four studies that were presented in the conference, machine learning technologies are used for data analysis and prediction (Haupt et al., 2018; Garcia-Marti et al., 2018; Bari, 2018; Schultz et al., 2018). Besides using standard meteorological datasets, these studies employed additional data to infer relationships that are relevant to the end user. For example, prediction of solar power output over a future time period requires the inclusion of historical and real-time solar energy production data (Haupt et al., 2018). It was observed that the use of machine learning approaches in weather and climate science is increasing. These approaches are powerful, for instance, in emulating processes that are not resolved in simulation models (because of computational costs), in calibrating or post-processing simulation results and in building models to describe or forecast meteorological and climatological events. The caveats, on the other hand, are that trained models are not transparent as models based on laws of physics and their results can be hard to interpret. Following the open science principle, machine learning approaches should be understandable and reusable by other researchers. Emerging fields like Explainable AI and knowledge based machine learning may provide approaches that help humans experts to understand how machine learning results are produced (Adadi and Berrada, 2018). Data-driven machine learning approaches should be combined with knowledge on physical processes (Dueben and Bauer, 2018; Reichstein et al., 2019) to gain further understanding of Earth system science problems. More broadly, machine learning methods should be accompanied by proper validation and verification.

This use of software, motivated by open science principles, requires a suitable digital infrastructure. The cloud appears to be a potential avenue as it enables individual researchers to gain access to high computing resources, vast amounts of storage and suites of software tools. In our session, three digital platforms were presented that use cloud technologies to create a virtual research environment in which scientific end-users can store, analyze and share their data (Ramamurthy, 2018; Hut et al., 2018; Bendoukha, 2018).The session participants also observed, however, that current platforms such as the Open Geospatial

Consortium (D. Maidment et al., 2011) and JRC Earth Observation Data and Processing Platform (Soille et al., 2017), do not seem to increase the extent of scientific collaboration, particularly across disciplines. This may be partly due to the fact that these platforms have each implemented their own set of standards both for data formats and interfaces to access these data. Since scientists are required to invest time and effort in working with a specific platform, this heterogeneity can pose obstacles to their collaboration with researchers on another platform.

## 5   DISCUSSION

This paper reflects the current discourse on open science in weather and climate research and the opportunities for sharing and combining data, software and infrastructure. Although this is an ongoing debate in the community, the individual aspects are usually discussed in isolation. Our approach in this paper takes the discourse further by focusing on 'open science in weather and climate research' as a whole, a concept which hardly received attention so far. We consider all aspects of open science,

among which compute infrastructures and stakeholders, and discuss the challenges and opportunities of recent open science developments in data, software and hardware. We are basing our claims on the insights and observations made during the conference session on "Weather and Climate Science in the Digital Era". These observations are representative of what we are seeing in the field, although we recognize that our analysis is not complete. However, we believe that, given our experience, we have a solid view of the accomplishments of open science along with what still needs to be implemented.

The studies presented in the session show the value of sharing open data, and using and developing open source software and open platforms. Scientific advances are shown, for instance, through combining data sets and including non-standard meteorological data such as social media posts and observations from amateur weather stations.The increase in accuracy and skill of forecasts at local scales show improved consistency of data products and improved efficiency and skill of simulations, often crossing different disciplines. The utilisation of machine learning and increased computational capabilities have facilitated the

use of disparate sources of data. In our conference session we concluded that sharing data and code offers many opportunities for scientific progress, leads to better reproducible science and vastly enhances the user base. However, we realized that open publication of data and code is not sufficient to achieve open weather and climate science and that there are important issues to address, which are described below.

    The findability and accessibility of data increasingly receives attention in weather and climate research, and common file

and metadata formats increase interoperability. However, for many data sets the implementation of the FAIR principles remains a challenge due to their origin, scalability issues or legal barriers. We also acknowledge that data quality can be difficult to

judge, depending on its intended use, or the reason for its generation. Addressing this data quality challenge requires continued discussion on what aspects of open data can be implemented generically and what aspects are specific.

Technologically, the promise of using modern digital technologies is not always met due to the complexity of software platforms. While this paper does not address hardware, this is true for hardware and the software run by these hardware as well. A further development of platforms should facilitate the ease-of-use and provenance. This also calls for more attention to research software engineering where collaboration and interaction between software engineers and domain researchers can lead to optimal use of open science tools and methods.

As mentioned before, open science concerns various stakeholders in addition to scholars. Data management and programming have become an integral part of current research practice, and these activities require specific digital skills (Akhmerov et al., 2019). It is therefore important to acknowledge and define roles, responsibilities and mandates concerning data stewardship and research software engineering. This requires institutional change as the personnel portfolio of academic institutions needs to become more diverse, and in addition, a broader consideration of the impact of academic work beyond scientific publications and teaching.

In order to remove legal boundaries on sharing data, it is important to also engage non-academic parties such as operational and commercial meteorological institutions in open science. New policies regarding open science should be developed in an inclusive way to engage all stakeholders. Open science strategies and policies facilitate a higher quality of scientific research, increased collaboration, and engagement between research and society, which in turn can lead to higher social and economic impacts of public research (OECD, 2015).

## 6 CONCLUSION AND RECOMMENDATIONS

Alongside the issues and challenges regarding open weather and climate science, this paper also discusses opportunities and possible solutions for these issues. We have compiled these into the following list of concrete recommendations which will bring us closer to open weather and climate science. Some of these recommendations are new, others are ongoing, but still hold.

Regarding data:

– **Developers should include post-processing steps in their simulation models. This requires additional compute and memory.**

This is ongoing work

– **Researchers using data from simulation models should increase parallelism in the data processing tool chain. This requires additional expertise in cloud computing, parallel and distributed computing.**

This is recent work, see e.g. (Team, 2016; Zaharia et al., 2016; Hoyer and Hamman, 2017; Office, 2010) for existing approaches

– **Individual researchers should be encouraged to publish scientific data in dedicated data journals.**

This is recent work, see e.g. Geoscience Data Journal, Scientific Data and Earth System Data

Regarding software and infrastructure:

- **Cloud technologies provide a suitable digital infrastructure for individual researchers to gain access to resources and tools and to collaborate with colleagues.**

  This is recent work, see e.g.(D. Maidment et al., 2011; Soille et al., 2017)

- **Platforms for scientific collaboration should consider interoperability and user friendliness.**

This is a new recommendation.

Regarding stakeholders and context:

- **Nondisclosure agreements should be signed between scientific partners and weather services and the latter should be allowed to act as trusted agents. This requires including trusted agents as new roles in the scientific process and engaging them as stakeholders in new policies regarding open weather and climate science.**

This is a new recommendation.

- **Funders should request researchers to adhere to FAIR principles.**

  This is recent work.

- **All stakeholders should acknowledge and define roles, responsibilities and mandates concerning data stewardship and research software engineering. This requires both institutional change and a broader consideration of the**

**impact of academic work.**

  This is a new recommendation.

Open science has implications for the stakeholders, the institutions and the system of science as a whole. It requires effort to change, but the benefits are large. Openly sharing data, code, and knowledge vastly enhances the user base, which means manifold growth of opportunities for new discoveries. As we observed from our conference session, this can lead to an improved

understanding of our complex world.

*Author contributions.* MGdV and WH organized the conference session and were lead writers of the manuscript. All authors contributed to the presentations and discussion in the conference session and to the writing of the manuscript.

*Competing interests.* The authors declare that they have no conflict of interest.

*Acknowledgements.* The authors would like to acknowledge both the Netherlands eScience Center and the program committee of the Weather
& Climate session for their organizational efforts. The session created a unique opportunity for specialists in the domain of weather and
climate science, data and computer scientists to exchange ideas and knowledge. AM and JB acknowledge the ESCAPE projects which
have received funding from the European Union's Horizon 2020 research and innovation programme under grant agreement No 671627
(ESCAPE) and No 800897 (ESCAPE2). SEH is with the National Center for Atmospheric Research in the U.S., which is a major facility
sponsored by the National Science Foundation under Cooperative Agreement No. 1852977.

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
