# Peer review of "Open Weather and Climate Science in the Digital Era"

_Geoscience Communication, 2019_

## Referee Comment (RC1) · Peter Düben (Referee) · 19 Nov 2019

The paper is an interesting read on an important and timely topic. The length is appropriate, and the authors represent a broad spectrum of subjects within Geosciences. However, the paper could improve significantly if the text would be more specific and if more specific examples would be provided for the claims in the text (see list of points below but there are many more places in the text). I think that it would make the paper much more credible if the authors would provide a list of action points to improve the situation at the end of the paper. However, I leave this for the authors to decide. The English should be improved.

**Specific comments:**

[Figure]

- ll.6-9: How is this shown?

- ll. 10-14: What is special about the origin, scalability and legal barriers? Why does the complexity limit collaboration? Can you give examples?

- l. 14: Why is there a need for new roles?

- l. 36: Was this really both short and long wave? If you refer to the 90s, you should also cite the original papers by Chevallier et al.

- l. 56: Lagging behind whom? Can you give an example?

- End of section 1: It would be good to give a hint about the structure of the following. The reader does not know what to expect from the rest of the paper.

- l. 71-74: Which or at least how many countries? How many Funders and Research institutes? Can you give examples?

- l.82: As you elaborate later on, OpenIFS is not Open Source as it has a (free) license.

- End of section 2: You could also mention Reanalysis data here.

- l.108: "clearly enrich their research" Can you give an example how?

- l. 114: What is "CF"?

- l. 124: What do you mean by "performance scalability". Software tools that allow to evaluate data at scale on supercomputers? How is data interoperable?

- l. 132: Which tools? Can you name them?

- l. 144: Which Journals? Can you name them?

- l. 150: Can you outline some of the examples in more detail?

- . "The studies show that use of machine learning methods has added value because models are built with data beyond standard meteorological data. For example, local conditions related to the natural and built environment that cannot be captured easily in simulation models can be taken into account through trained models." I do not understand this. Can you rephrase?

- l. 177: Can you name examples for hardware and software platforms. And can you define what you mean by "platform" in this context?

- "data such as that of the environment and citizen science sources." I do not know which data sets you are referring to here.

- "The increase in accuracy and skill of forecasts at local scales are shown, improved consistency of data products and improved efficiency and skill of simulations, often crossing different disciplines." Again, I do not understand this. Do you mean "show" instead of "are shown,"?

- l. 194: Which issues?

- "Technologically, the promise of using modern digital technologies is not always met due to the complexity of software platforms." I do not understand this.

**Minor points:**

- l. 9: Rephrase: "that here are"

- l. 32: Rephrase "since ensured"

- l.45: Rephrase: "use of using"

- l. 171: "Emerging fields"

- l. 192: "science and it vastly" -> "science, and vastly"

- l. 206: "science science"

- l. 213: Rephrase "are a means to"

- l. 216: Rephrase "openly vastly"

---

## Short Comment (SC1) · 10 Jan 2020

APARNA RADHAKRISHNAN

aparna.radhakrishnan@princeton.edu

General comments:

This paper is indeed interesting and awakening. This statement from the paper does have quite an impact. "As can already be observed from the studies presented in the conference it leads to much faster progress in understanding the world."

Thanks for your contributions.

Specific comments:

Throughout this review P<N>, L<N> represents the page number and line number respectively.

[Figure]

P2, L 45. There is a line that talks about the "third development". The construction of this paragraph could be slightly modified to explicitly present the three developments, for a better flow.

P3, L 63. Section 2, Consider eliminating too many "and" conjunctions.

P4, L 94-96. Examples or relevant references cited will improve the effectiveness of this statement.

P4, L 106 onwards. Some parts in 3.1 Open Data seem to fall under 3.2 Open software. But, this could also mean they are very coupled. No changes necessarily needed here.

P5, L 118. While interpreting , "Making data and software findable..", software may include tools that lead to the data. I think some level of paraphrasing may be required in this paragraph to make the message from the paper more evident, about making all the components adhere to FAIR goal as a whole.

P5, L 126. This paragraph does provide good insights. But, the final message is not translated well enough as to how this affects open data/science.

P5, L 131. Just a note- Removing the need for post-processing by incorporating as many steps as possible within the model itself can make the model computationally even more expensive. Thus, when there is a use-case to share model source code, one may still find it challenging, though open. Though there is one helpful cloud computation reference cited, I would have expected to see more bits about cloud computing in this paper, in this particular section.

P6, L 161. Punctuation. Add comma after conference.

P6, L 178 The message/action item here seems to have not translated well here. It does sound contradictory, but the essence of the message might be lost, regarding the technical challenges and reduced scope for multi-discipline collaboration. Please paraphrase this to improve the paragraph.

P7, L 194 Punctuation. Replace "here" with "there.

P7, L 216 This statement is well put in terms of sharing knowledge. I hope this can be reflected more in the paper.

---

## Referee Comment (RC2) · John K. Hillier (Referee) · 16 Jan 2020

I would like to say 'moderate' revisions, but this options is not available in the tick-boxes. Please see attachment for Review.

Please also note the supplement to this comment:
https://www.geosci-commun-discuss.net/gc-2019-22/gc-2019-22-RC2-supplement.pdf
* * *

---

## Author Comment (AC1) · 2 Mar 2020

Please see the attached file for our reply to the reviewer's comments

Please also note the supplement to this comment:
https://www.geosci-commun-discuss.net/gc-2019-22/gc-2019-22-AC1-supplement.pdf

---

## Author Response (AR1)

**Authors' reply to reviewer comments  GC-2019-22**

Weather and Climate Science in the Digital Era, Martine G. de Vos et al.

**Reviewer Comment 1**

**General:**

*The paper is an interesting read on an important and timely topic. The length is appropriate, and the authors represent a broad spectrum of subjects within Geosciences. However, the paper could improve significantly if the text would be more specific and if more specific examples would be provided for the claims in the text (see list of points below but there are many more places in the text).*

We agree with the reviewer that we could be more specific on how we have collected our observations and how these support the claims in the text. We will add a dedicated 'Methods' section (see below for the suggested text). Besides, throughout the paper we will rephrase text to be more specific on our observations and how these support our story.

" **Methods :** The focus of the conference session was on data and compute intensive approaches that are applied in weather and climate science. The session comprised 10 oral abstract presentations,
one keynote talk, and 6 short poster pitches. The 16 participants were either presenters or involved in the organization of the session, and represented domain science, as well as computer and data sciences.
The first part of the session was dedicated to the presentations.The second part was interactive. In three groups of each 5 or 6 persons the participants discussed
the "challenges and opportunities regarding open weather and climate science" and noted their findings on a flipchart. The findings of each group were presented and discussed in a following plenary session. Observations and insights from the plenary discussion were documented.
The observations in this paper are based on both the insights from the studies presented in the session, and the notes made during the interactive part of the session. The majority of the participants from the session also contributed to this paper. As such this paper represents a shared view of the participants, i.e., a group of experts in weather and climate science, on the digital and open science developments in their field."

*I think that it would make the paper much more credible if the authors would provide a list of action points to improve the situation at the end of the paper. However, I leave this for the authors to decide.*

We thank the reviewer for this great suggestion. At the end of the paper, we will provide a list of action points or conclusions that are described in the different sections of the paper.

*The English should be improved.*

We agree with the reviewer and get a native speaker to edit the manuscript

**Specific comments:**

*• ll.6-9: How is this shown?*

We will rephrase the paragraph to clarify its meaning and add concrete examples that illustrate the importance of shared data and software:

"The majority of studies (roughly 80 %) presented in the conference session depended in some way or another on shared data and software. For example, many studies included open datasets from disparate sources to improve accuracy of forecasts on the local scale, or to extend analyses beyond the domain of weather and climate. Furthermore, shared software is a prerequisite for the studies that presented systems like a model coupling framework or a digital collaboration platform. Although these studies showed that sharing code and data is important, the consensus among the participants was that this is not sufficient to achieve open weather and climate science and that there are important issues to address."

*• ll. 10-14: What is special about the origin, scalability and legal barriers?*

For instance, many data sources come from private industry who may see a competitive advantage to maintaining privacy. But those data may prove useful to the weather community for improving initial conditions of forecast models. Such corundums may be solved by signing nondisclosure agreements and allow weather service to act as trusted agents who use the data for the public good without disclosing their details.

We will include this explanation in the abstract and in the corresponding sections.

*• ll. 10-14: Why does the complexity limit collaboration? Can you give examples?*

We will elaborate the text in the abstract and the corresponding section. Please see the last comment on software platforms for the text suggestion.

*• l. 14: Why is there a need for new roles?*

Data management and programming have become an integral part of current research practice, and these activities require specific digital skills. It is therefore important to acknowledge and

define roles, responsibilities and mandates concerning data stewardship and research software engineering.
The aforementioned trusted agents can also be considered a new role
We will include this explanation in the corresponding sections.

*• l. 36: Was this really both short and long wave? If you refer to the 90s, you should also cite the original papers by Chevallier et al.*
We can confirm that neural nets have been used for both short and long wave radiation. We will rephrase the sentence and add the corresponding references.

*• l. 56: Lagging behind whom? Can you give an example?*
The reviewer rightly points out that it is not clear who, or which field we compare to. In fact, open sharing of data, software and vocabularies is only true common practice in a few fields such as astronomy and genomics. Most scientific fields, including weather and climate science, can be considered lagging behind. Furthermore, the actual point was to show that these weather and climate science are mature in terms of applying digital technologies, while the implementation of open science methodologies is less advanced

We will rephrase the corresponding paragraph accordingly.

*• End of section 1: It would be good to give a hint about the structure of the following. The reader does not know what to expect from the rest of the paper.*
We adopt the advice of the reviewer and will clarify the structure of the rest of the paper at the end of the introduction section

*• l. 71-74: Which or at least how many countries? How many Funders and Research institutes? Can you give examples?*
We will try to be more specific and add examples and references to this paragraph.
"Europe and the United States have made efforts to adapt legal frameworks and implement policy initiatives greater openness in scientific research (OECD, 2015; National Science Foundation, 2018). Several countries provide digital infrastructure based on rich metadata for the resources in the research environment, that support their optimal re-use (Mons, 2017). Examples include the European Open Science Cloud in Europe (Directorate-General,2018) , NIH Data Commons projects in the United States, AARnet in Australia (AARNet, 2018) and the African Data Intensive Research Cloud in South Africa (Simmonds,2016). Funders and research institute have announced policies encouraging, mandating, or specifically financing open research practices (Mckiernan et al.,2016; Wilkinson et al, 2016). Examples include the National Science Foundation (NSF) in the United States (National Science Board,2011), CERN in Switzerland (CERN, 2014), the Netherlands Organization for Scientific Research (NWO) (NWO, 2019) and the United Nations Educational, Scientific and Cultural Organization (UNESCO) (Unesco, 2013)."

Unfortunately, we are not able to provide quantitative information.

Mons, B., Neylon, C., Velterop, J., Dumontier, M., Da Silva Santos, L. O. B., & Wilkinson, M. D. (2017). Cloudy, increasingly FAIR; Revisiting the FAIR Data guiding principles for the European Open Science Cloud. Information Services and Use, 37(1), 49–56. https://doi.org/10.3233/ISU-170824

Directorate-General for Research and Innovation. (2018). Prompting an EOSC in practice. Final report and recommendations of the Commission 2nd High Level Expert Group on the European Open Science Cloud (EOSC). https://doi.org/10.2777/112658

AARNet. (2018). ANNUAL REPORT / 2018 DATA CONNECTOR FOR THE FUTURE. Chatswood, Australia.

CERN-OPEN-2014-049. (2014). Open Access Policy for CERN Physics Publication.

R. Simmonds, Taylor, R., Horrell, J., Fanaroff, B., Sithole, H., Rensburg, S. J. van, & Al., E. (2016). The African data intensive research cloud. IST - Africa Week Conference.

National Science Board. (2011). Digital Research Data Sharing and Management.

NWO executive board. (2019). Connecting Science and Society - NWO strategy 2019-2022.

UNESCO Executive board. (2013). Open Access Policy concerning UNESCO publications.

*• l.82: As you elaborate later on, OpenIFS is not Open Source as it has a (free) license.*
The reviewer is right. We will rephrase the sentence.

*• End of section 2: You could also mention Reanalysis data here.*
We adopt the reviewers suggestion and will add the following text:
"Already since the 1990s the international meteorological and climate research communities started sharing data.  Examples of data sharing with common file and metadata formats are reanalysis data, starting with NCEP/NCAR reanalysis and ECMWFs ERA reanalysis data products (e.g. Dee et al 2011, Kalnay et al. 1996) and coupled model intercomparison projects (Taylor et al. 2012)."

*• l.108: "clearly enrich their research" Can you give an example how?*
Examples include the various reanalysis datasets published by the ECMWF and NOAA/NCAR that are made freely available to the community and application of the open models, such as WRF.
We will add these examples to the section.

*• l. 114: What is "CF"?*
CF conventions provide guidelines for the use of metadata in the netCDF file. We will rephrase the paragraph and include the meaning and use of CF

*• l. 124: What do you mean by "performance scalability". Software tools that allow to evaluate data at scale on supercomputers? How is data interoperable?*

*• l. 132: Which tools? Can you name them?*
We will rephrase the corresponding sentences to clarify the challenges of producing FAIR weather and climate model data:
"Regarding open and interoperable weather and climate model data, i.e. data and metadata that are formatted according to community standards (CF, CMIP, WMO), we consider performance scalability as the foremost technological challenge. Whereas high-resolution weather and climate data is predominantly produced on large clusters using many compute nodes, subsequent data processing and analysis is often still confined to a single CPU, and hence does not scale easily with, e.g., increased model resolution. Producing FAIR model data via traditional post-processing pipelines is quickly becoming unfeasible for high-resolution climate model data due to the sheer volume and complexity of the model output as noted above."

*• l. 144: Which Journals? Can you name them?*
We will add some examples to the text:
"Data journals, like Geoscience data journal (Royal Meteorological Society), Scientific Data (Springer Nature) and Earth System Data (Copernicus Publications), are a partial remedy, as these provide open access platforms where scientific data can be peer-reviewed and formally published."

*• l. 150: Can you outline some of the examples in more detail?*
We will elaborate the examples and rephrase the paragraph as follows:
"The conference session provided excellent examples of tools and approaches that were developed and made openly available to the research community. For example, approaches to reduce the computational or post processing costs of existing simulation models (Stringer et al., 2018; Behrens et al., 2018; van den Oord et al., 2018, Jansson et al., 2018) and approaches to integrate data sets from different sources (van Haren et al., 2018; Schultz et al., 2018). Several of the studies in the session presented an approach for which open data and software is a prerequisite, for example because these comprise a model coupling framework or a digital collaboration platform (Pelupessy et al., 2018; Ramamurthy, 2018; Hut et al., 2018; Bendoukha, 2018)."

*• . "The studies show that use of machine learning methods has added value because models are built with data beyond standard meteorological data. For example, local conditions related to the natural and built environment that cannot be captured easily in simulation models can be taken into account through trained models." I do not understand this. Can you rephrase?*
This paragraph is about the use of data beyond the standard meteorological datasets. We will rephrase the paragraph to clarify this.

*• l. 177: Can you name examples for hardware and software platforms. And can you define what you mean by "platform" in this context?*
These platforms refer to digital platforms that use cloud technologies to create a virtual research

environment where scientific end-users can store, analyze and share their data. In the conference session several of these platforms were presented. An example of a current platform is the Open geospatial Consortium. We will rephrase the paragraph to clarify this.

• *"data such as that of the environment and citizen science sources." I do not know which data sets you are referring to here.*
This sentence is referring to the data sets described in the section on open data, i.e., social media posts and observations from amateur weather stations. We will rephrase the sentence to make this clear

• *"The increase in accuracy and skill of forecasts at local scales are shown, improved consistency of data products and improved efficiency and skill of simulations, often crossing different disciplines." Again, I do not understand this. Do you mean "show" instead of "are shown,"?*
The reviewer is right, it should have been "show". We will rephrase the sentence accordingly.

• *l. 194: Which issues?*
This term refers to the issues described in the next paragraphs in the same section. The reviewer rightly points out that this should be clear from the text. We will rephrase the text in the section correspondingly.

• *"Technologically, the promise of using modern digital technologies is not always met due to the complexity of software platforms." I do not understand this.*
The cloud appears to be a potential avenue, as it enables individual researchers to gain access to high computing resources, vast amounts of storage and a suite of software tools. In our session, several digital platforms were presented, that use cloud technologies to create a virtual research environment where scientific end-users can store, analyze and share their data. The participants also observed, however, that current platforms, like the Open Geospatial Consortium and JRC Earth Observation Data and Processing Platform, do not seem to increase the extent of scientific collaboration, especially across disciplines. This may be partly due to the fact that these platforms each have implemented their own set of standards for both data formats and interfaces to access these data. Since scientists are required to invest time and effort in working with a specific platform, the heterogeneity poses hurdles to their collaboration with researchers on another platform.

We will rephrase the paragraph to clarify this:

**Minor points:**

• *l. 9: Rephrase: "that here are"*

We will rephrase the sentence

*• l. 32: Rephrase "since ensured"*
We will rephrase the sentence

*• l.45: Rephrase: "use of using"*
We will rephrase the sentence

**Reviewer Comment 2**

**Major Points**

*1. There are a number of typographical mistakes, albeit mainly subtle. So please get a native english speaker to proof-read the manuscript. Namely, I have not attempted to pick up all typos.*
We adopt the advice of the reviewer and get a native speaker to edit the manuscript

*2. The methodology (i.e. what was done in the session) needs to be clarified e.g. (i) were specific questions/topics posed for this research exercise [which it was], (ii) elicitation by sticky notes or hands in the air or by the co-authors making notes of what the group said? I think the observational data are (i) L57-558 - a specific session to discuss (by unstated means) the issues (unspecified in detail), and (ii) L20 insights from the work in the rest of the conference (by unstated means). A 'Methods' section needs to be added, which is one place where the questions asked at the session could be stated.*
*3. The 'novelty' (i.e. what is reported here that is not stated elsewhere) is difficult to distinguish, although a Methods section and taking care to phrase the results/discussion in terms of the evidential basis of insights should fix this.*
*4. The Abstract portrays all the thoughts as entirely new, rather than emerging from a context. e.g. L8 'we observed' - we reaffirm? we agree with the informal subject-wide consensus? Please rephrase where appropriate. As an editor of GC, I note that this was submitted as a review article, but it may be better classified as a standard paper.*
The approach followed in the session was similar to a 'focus group' approach where experts in share views and experiences. This paper is not a classical science paper addressing a well posed problem, but synthesizes those experiences from arguably a wide range of specialists. We agree with the reviewer that both the context and type of this research, and the methodology deserve clarification. We adopt the advice of adding a dedicated 'Methods' section (see below for the suggested text). Besides, throughout the paper we will rephrase text to correctly reflect our methodology.

" **Methods :** The focus of the conference session was on data and compute intensive approaches that are applied in weather and climate science. The session comprised 10 oral abstract presentations,
one keynote talk, and 6 short poster pitches. The 16 participants were either presenters or involved in the organization of the session, and represented domain science, as well as computer and data sciences.
The first part of the session was dedicated to the presentations.The second part was interactive. In three groups of each 5 or 6 persons the participants discussed
the "challenges and opportunities regarding open weather and climate science" and noted their findings on a flipchart. The findings of each group were presented and discussed in a following plenary session. Observations and insights from the plenary discussion were documented.
The observations in this paper are based on both the insights from the studies presented in the session, and the notes made during the interactive part of the session. The majority of the participants from the session also contributed to this paper. As such this paper represents a shared view of the participants, i.e., a group of experts in weather and climate science, on the digital and open science developments in their field."

**Minor Points**

*Title - The paper's contents are about open access, not digital (see L2&3). Suggest changing title to reflect this.*
We agree with the reviewer that this paper is about open science. In fact, we think we really do include both open science and the digital era. We suggest that we include both terms in the title, i.e., Open Weather and Climate Science in the Digital Era. In the introduction we will point out what we mean by "digital era".

*L6 - 'the studies in the conference session showed' - How exactly?*
We will rephrase the paragraph to clarify its meaning and add concrete examples that illustrate the importance of shared data and software:
"The majority of studies (roughly 80 %) presented in the conference session depended in some way or another on shared data and software. For example, many studies included open datasets from disparate sources to improve accuracy of forecasts on the local scale, or to extend analyses beyond the domain of weather and climate. Furthermore, shared software is a prerequisite for the studies that presented systems like a model coupling framework or a digital collaboration platform. Although these studies showed that sharing code and data is important, the consensus among the participants was that this is not sufficient to achieve open weather and climate science and that there are important issues to address."

*L8 - 'we observed' - how (in)formally was this done?*

*L62 - A brief comment on the limitations/benefits of the approach used to bring together the information for this paper appears necessary in the Methods section.*
*L99 & 103 - Session/sessions? One 'session' with multiple time blocks?*
*L103 - A hint of what was done. Good, but please expand in a Methods section. Using the standard Method/Results/Discussion format might help the clarity of the work. Having everything merged into thematic section currently makes determining what this paper adds difficult, although by clearly stating which evidence comes from where and moving from data to discussion within the existing sections might also work.*
*L104 - 'Discussed'. Please elaborate. e.g. who is 'we'. The co-authors of this paper? How was it determined what are 'common findings' and 'highlights'?*
*L118 - Example of where evidential basis could be clarified. 'we recognized': we as co-authors discussing and concluding, we in the session, and how was this recognized (e.g. large majority in room, or someone mentioned, or did all participants agree to a circulated notes/minutes?).*
We agree with the reviewer that both the context and type of this research, and the methodology deserve clarification. We adopt the advice of adding a dedicated 'Methods' section (see our reply in 'major points' for the suggested text). Besides, throughout the paper we will rephrase text to correctly reflect our methodology.

*L9 - Typo - 'there' not here*
We will rephrase the sentence

*L11 - 'primarily due to'? i.e. either these were refined from a list for some reason, or is this the complete list of possibilities?*
This statement refers to the section where these barriers are described in more detail. For instance, many data sources come from private industry who may see a competitive advantage to maintaining privacy. But those data may prove useful to the weather community for improving initial conditions of forecast models. Such corundums may be solved by signing nondisclosure agreements and allow weather service to act as trusted agents who use the data for the public good without disclosing their details.

We will include this explanation in the abstract and in the corresponding section.

*L19-20 - It is claimed that 'much faster progress' is being made as 'observed from the studies presented in the conference'. This is quite a leap of logic, and is one illustration of how the manuscript could be more tightly argued and/or presented. If this is simply the authors impression, this is fine, but should be clarified by adding 'we believe' or similar. If written as a statement, and evidential basis should be provided in the new data collected. If this is simply a confirmation of what is in the existing literature (i.e. L52-53) then this should be also clarified.*
The reviewer rightly points out that this is the authors' view. We will rephrase the sentence accordingly.

*L21 - Typo - .. computationally intensive ...*

We will rephrase the sentence

*L22 - Introduction. A wide range of topics and issues are introduced here. They are placed in historical context, which is good. But, the treatment of these becomes quite vague when the actual session is mentioned (L58- 59)*

The introduction discusses the role and use of technology in weather and climate science in history as well as the 'digital era '. We will clarify this as mentioned in the reply on the first comment.

We will move the description of the session to the new Methods section.

*L39-48 - This paragraph is currently un-referenced. Please add these.*

We adopt the reviewers suggestion and will add the following references to the paragraph:

Bauer, P., Thorpe, A., & Brunet, G. (2015). The quiet revolution of numerical weather prediction. Nature, 525(7567), 47–55. https://doi.org/10.1038/nature14956

Huntingford, C., Jeffers, E. S., Bonsall, M. B., Christensen, H. M., Lees, T., & Yang, H. (2019). Machine learning and artificial intelligence to aid climate change research and preparedness. Environmental Research Letters, 14(12), 124007. https://doi.org/10.1088/1748-9326/ab4e55

Schneider, T., Lan, S., Stuart, A., & Teixeira, J. (2017). Earth System Modeling 2.0: A Blueprint for Models That Learn From Observations and Targeted High-Resolution Simulations. Geophysical Research Letters, 44(24), 12,396-12,417. https://doi.org/10.1002/2017GL076101

Reichstein, M., Camps-Valls, G., Stevens, B., Jung, M., Denzler, J., Carvalhais, N., & Prabhat. (2019). Deep learning and process understanding for data-driven Earth system science. Nature, 566(7743), 195–204. https://doi.org/10.1038/s41586-019-0912-1

Ruti, P., Tarasova, O., Keller, J., Carmichael, G., Hov, Ø., Jones, S., … Yamaji, M. (2019). Advancing Research for Seamless Earth System Prediction. Bulletin of the American Meteorological Society, (August 2019), 23–35. https://doi.org/10.1175/bams-d-17-0302.1

L40 - 'exascale' - I don't know this word. Please add a reference or two so that non-specialists can inform themselves.

The term 'exascale' computing refers to $10^{18}$ operations per second, a factor of 1000 beyond current machines.

We will explained the term in the text and add a reference to the sentence:

Reed, D. A., & Dongarra, J. (2015). Exascale computing and big data. Communications of the ACM, 58(7), 56–68. https://doi.org/10.1145/2699414

*L62 - Open science. This appears to be a literature review, unrelated to the session mentioned. Was the session simply used as a brainstorming exercise to get the information together for such literature reviews? If so, again this is fine, but include a Methods section to state this, even if it's only a paragraph long. When the paper is revised, I would expect to distinguish whether the information is (i) in the literature, and being brought together here (ii) views of people in the room etc ...... And, this will allow the contribution of this paper to be clarified/determined. If this section is a review, say 'review' not 'explore', but my Methods points still stand w.r.t later sections.*

The reviewer rightly points out that this section contains a literature review on open science. We will clarify this both in this section, i.e., rephrase the 'explore' sentence, and at the end of the introduction section, where we explain the structure of the paper. We will also add a dedicated 'Methods' section (see our reply in 'major points' for the suggested text).

*L106 - Please try to be specific. Does 'many' mean 5, 50% or something different? It should be possible to give numbers for papers in your session, or you might randomly sample the conference in a desk-based exercise.*

We agree with the reviewer and throughout the paper we will rephrase text to be more specific on our method of data collection.

*General - Is there scope for a table of key points, or graphic to present the most important findings? I am a bit ambivalent about saying this as us readers shouldn't be lazy, but this could usefully highlight the key detailed points. Example of how this could be done - each co-author gets 3 votes, and size of coloured blob relates to number of votes in the graphic.*

We thank the reviewer for this great suggestion. At the end of the paper, we will provide a list of action points or conclusions that are described in the different sections of the paper.

**Short Comment 1**

*P2, L 45. There is a line that talks about the "third development". The construction of this paragraph could be slightly modified to explicitly present the three developments, for a better flow.*

We adopt the suggestion of the reviewer and will modify the construction of the paragraph

*P3, L 63. Section 2, Consider eliminating too many "and" conjunctions.*

We adopt the suggestion of the reviewer and will check the text for unnecessary "and" conjunctions

*P4, L 94-96. Examples or relevant references cited will improve the effectiveness of this statement.*

In fact, open sharing of data, software and vocabularies is only true common practice in a few fields such as astronomy and genomics. Most scientific fields, including weather and climate science, can be considered lagging behind. We will add a few references to support this.

*P4, L 106 onwards. Some parts in 3.1 Open Data seem to fall under 3.2 Open software. But, this could also mean they are very coupled. No changes necessarily needed here.*

*P5, L 118. While interpreting , "Making data and software findable..", software may include tools that lead to the data. I think some level of paraphrasing may be required in this paragraph to make the message from the paper more evident, about making all the components adhere to FAIR goal as a whole.*
The reviewer is right, data and software in are connected and both should adhere to the FAIR principles. We will modify the text of this paragraph (and if necessary other parts of the paper) to clarify this message.

P5, L 126. *This paragraph does provide good insights. But, the final message is not translated well enough as to how this affects open data/science.*
*P5, L 131. Just a note- Removing the need for post-processing by incorporating as many steps as possible within the model itself can make the model computationally even more expensive. Thus, when there is a use-case to share model source code, one may still find it challenging, though open. Though there is one helpful cloud computation reference cited, I would have expected to see more bits about cloud computing in this paper, in this particular section.*
We agree with the reviewer that the impact on open data/science can be stated more clearly. We included a more elaborate description that producing FAIR model data is necessary, but can not be achieved through traditional post-processing pipelines.
Furthermore, we agree with the reviewer that cloud computing technologies, like xarray, Dask, and Apache SPARK, could be useful, since data processing and analysis pipelines usually do not require communication between parallel jobs. One of the key aspects, however, is the capability of the developer, usually a meteorologist or climate scientist, to adopt a new programming paradigm that allows the parallel execution of the workflow on cloud infrastructure. Here research software engineers may play a key role by, e.g., building useful tooling on top of existing low-level platforms like Apache Spark or Dask.

We will rephrase the paragraph accordingly.

*P6, L 161. Punctuation. Add comma after conference.*
We will rephrase the sentence

*P6, L 178 The message/action item here seems to have not translated well here. It does sound contradictory, but the essence of the message might be lost, regarding the technical challenges*

*and reduced scope for multi-discipline collaboration. Please paraphrase this to improve the paragraph.*

We will rephrase the paragraph to clarify the message:

"The use of software as presented above, motivated by open science principles, requires a suitable digital infrastructure. The cloud appears to be a potential avenue, as it enables individual researchers to gain access to high computing resources, vast amounts of storage and a suite of software tools. In our session, several digital platforms were presented, that use cloud technologies to create a virtual research environment where scientific end-users can store, analyze and share their data. The participants also observed, however, that current platforms, like the Open Geospatial Consortium and JRC Earth Observation Data and Processing Platform, do not seem to increase the extent of scientific collaboration, especially across disciplines. This may be partly due to the fact that these platforms each have implemented their own set of standards for both data formats and interfaces to access these data. Since scientists are required to invest time and effort in working with a specific platform, the heterogeneity poses hurdles to their collaboration with researchers on another platform."

P7, L 194 Punctuation. Replace "here" with "there.
*We will rephrase the sentence*

*P7, L 216 This statement is well put in terms of sharing knowledge. I hope this can be reflected more in the paper.*

We thank the reviewer for this comment. Throughout the paper we will rephrase text to be more specific on our observations and how these support our story. At the end of the paper, we will compile a list of action points or conclusions, i.e., to improve the current situation, that are described in the different sections of the paper.

[revised manuscript text omitted]

CEDA Archive. The Natural Environment Research Council's Data Repository for Atmospheric Science and Earth Observation. Retrieved from http://archive.ceda.ac.uk/

Charney, J. G., FjÖrtoft, R., and Neumann, J. V.: Numerical Integration of the Barotropic Vorticity Equation, Tellus, 2, 237–254,https://doi.org/10.3402/tellusa.v2i4.8607, 1950.

Chevallier F, Chéruy F, Scott NA, Chedin A (1998) A neural network approach for a fast and accurate computation of longwave radiative budget. J Appl Meteorol 37:1385–1397

Copernicus Publications. Earth System Science Data. Retrieved from http://earth-system-science-data.net/

CONP. 2018. Canadian open neuroscience platform—a partnership with Brain Canada and Health Canada. Canadian Open Neuroscience Platform [online]: Available from conp.ca/.

Dask Development Team (2016). Dask: Library for dynamic task scheduling. URL https://dask.org

Dee, D. P., Uppala, S. M., Simmons, A. J., Berrisford, P., Poli, P., Kobayashi, S., Andrae, U., Balmaseda, M. A., Balsamo, G., Bauer,P., Bechtold, P., Beljaars, A. C., van de Berg, L., Bidlot, J., Bormann, N., Delsol, C., Dragani, R., Fuentes, M., Geer, A. J., Haim-berger, L., Healy, S. B., Hersbach, H., Hólm, E. V., Isaksen, L., Kållberg, P., Köhler, M., Matricardi, M., Mcnally, A. P., Monge-Sanz,B. M., Morcrette, J. J., Park, B. K., Peubey, C., de Rosnay, P., Tavolato, C., Thépaut, J. N., and Vitart, F.: The ERA-Interim reanalysis:Configuration and performance of the data assimilation system, Quarterly Journal of the Royal Meteorological Society, 137, 553–597,https://doi.org/10.1002/qj.828, 2011.

Directorate-General for Research and Innovation. (2018). Prompting an EOSC in practice. Final report and recommendations of the Commission 2nd High Level Expert Group on the European Open Science Cloud (EOSC). https://doi.org/10.2777/112658

Dueben, P. D. and Bauer, P.: Challenges and design choices for global weather and climate models based on machine learning, Geoscientific Model Development, 11, 3999–4009, https://doi.org/10.5194/gmd-11-3999-2018, 2018.

Easterbrook, S. M., & Johns, T. C. (2009). Engineering the software for understanding climate change. Computing in Science and Engineering, 11(6), 64–74. https://doi.org/10.1109/MCSE.2009.193

ESRI Inc. Geoportal XML Editor. Retrieved from https://github.com/Esri/geoportal-server/wiki/Geoportal-XML-Editor

Eyring, V., Bony, S., Meehl, G. A., Senior, C. A., Stouffer, R. J., and Taylor, K. E.: Overview of the Coupled MOdel Intercomparison ProjectPhase 6 (CMIP6) Experimental Design and Organization, Geoscientific Model Development, 9, https://doi.org/10.5194/gmd-9-1937-2016, 2016.

Fecher, B. and Friesike, S.: Open Science: One Term, Five Schools of Thought, in: Opening Science, edited by Bartling, S. and Friesike, S.,1, pp. 1–7, The Author(s), https://doi.org/10.1007/978-3-319-00026-8_2, 2014.2609

Fowler, M., & Highsmith, J. (2001). The Agile manifesto. Software Development, 9(8), 28–35.

Garcia-Marti, I., Noteboom, J. W., and Diks, P.: Detecting probability of ice formation on overhead lines of the Dutch railway network, in: IEEE 14th International Conference on e-Science, https://doi.org/10.1109/eScience.2018.00050, 2018.

Gene Ontology Consortium (2004). The Gene Ontology (GO) database and informatics resource. 32, 258–261. https://doi.org/10.1093/nar/gkh036

Gil, Y., David, C. H., Demir, I., Essawy, B. T., Fulweiler, R. W., Goodall, J. L., Karlstrom, L., Lee, H., Mills, H. J., Oh, J. H., Pierce,S. A., Pope, A., Tzeng, M. W., Villamizar, S. R., and Yu, X.: Toward the Geoscience Paper of the Future: Best practices for documenting and sharing research from data to software to provenance, Earth and Space Science, 3, 388–415, https://doi.org/10.1002/2015EA000136,2652016.

Granados Moreno P, Ali-Khan SE, Capps B, Caulfield T, Chalaud D, Edwards A, Gold ER, Rahimzadeh V, Thorogood A, Auld D, Bertier G, Breden F, Caron R, César PMDG, Cook-Deegan R, Doerr M, Duncan R, Issa AM, Reichman J, Simard J, So D, Vanamala S, and Joly Y. 2018. Open science precision medicine in Canada: Points to consider. FACETS 4: 1–19. doi:10.1139/Facets-2018-0034

[revised manuscript text omitted]

Shamir, L., Wallin, J. F., Allen, A., Berriman, B., Teuben, P., Robert J. Nemiroff, J. M., … DuPrie, K. (2013). Practices in source code sharing in astrophysics. Astronomy and Computing, 1, 54-58.

R. Simmonds et al., "The African Data Intensive Research Cloud," 2016 IST-Africa Week Conference, Durban, 2016, pp. 1-8.

Skamarock, W. C., Klemp, J. B., Dudhia, J., Gill, D. O., Liu, Z., Berner, J., Wang, W., Powers, J. G., Duda, M. G., Barker, D. M., andHuang, X.-Y.: A Description of the Advanced Research WRF Version 4. NCAR Tech. Note NCAR/TN-556+STR, Tech. rep., NCAR,https://doi.org/10.5065/1dfh-6p97, 2019.

Sletholt, M. T., Hannay, J. E., Pfahl, D., & Langtangen, H. P. (2012). What do we know about scientific software development's agile practices? Computing in Science and Engineering, 14(2), 24–36. https://doi.org/10.1109/MCSE.2011.113

Springer Nature. Scientific Data. Retrieved from http://www.nature.com/scientificdata/

Stringer, M., Jones, C., Hill, R., Dalvi, M., Johnson, C., and Walton, J.: A Hybrid-Resolution Earth System Model, in: IEEE 14th International Conference on e-Science, https://doi.org/10.1109/eScience.2018.00042, 2018.

Soille, P. &, Burger, A. &, Hasenohr, P. &, Kempeneers, Pieter & Rodriguez Aseretto, Dario & Syrris, V. &, Vasilev, V. &, & Marchi, D. (2017). THE JRC EARTH OBSERVATION DATA AND PROCESSING PLATFORM. Big Data From Space. Toulouse, France.

Sylla, M. B.: Review of meteorological / climate data sharing policy ( WMO Resolution 40 ) to promote their use to support Climate Information Services uptake in the African continent, in: Expert Group Meeting on data sharing policy in Africa, July, pp. 10–11, Dakar,Senegal, 2018.

Taylor, K. E., Stouffer, R. J., and Meehl, G. A.: An overview of CMIP5 and the experiment design, Bulletin of the American MeteorologicalSociety, 93, 485–498, https://doi.org/10.1175/BAMS-D-11-00094.1, 2012.

USGS Science Data Catalog (SDC). Retrieved from https://data.usgs.gov/datacatalog

van den Oord, G., Yepes, X., and Acosta, M.: Post-processing strategies for the ECMWF model, in: IEEE 14th International Conference on320e-Science, 2018.

van Haren, R., Koopmans, S., Steeneveld, G.-J., Theeuwes, N., Uijlenhoet, R., and Holtslag, A. A. M.: Weather reanalysis on an urban scale using WRF, in: IEEE 14th International Conference on e-Science, https://doi.org/10.1109/eScience.2018.00049, http://www2.mmm.ucar.edu/wrf/users/docs/arw{_}v3.pdf, 2018.

Wilkinson et al, M. D.: Comment: The FAIR Guiding Principles for scientific data management and stewardship, Scientific data, 3,325https://doi.org/DOI: 10.1038/sdata.2016.18, 2016.

Zaharia, M., Xin, R.S., Wendell, P., Das, T., Armbrust, M., Ankur, D., Xiangrui M., Rosen, J., Venkataraman, S., Franklin, M.J., Ghodsi, A., Gonzalez, J., Shenker, S., and Stoica., I. 2016. Apache Spark: a unified engine for big data processing. Commun. ACM 59, 11 (October 2016), 56–65. DOI:https://doi.org/10.1145/2934664

---

## Referee Report (RR1)

Weather and Climate Science in the Digital Era

Vos et al.

**Summary**

Firstly, I would like to comment that the authors' response is not particularly kind to reviewers in our attempt to determine if the requested changes have been made. Whilst areas of change have been highlighted, none of the responses are linked to the changes (e.g. line numbers in new document), and my major comments 2-4 with a single comment that is essentially *'Read the new Methods section, and we've made a selection of changes throughout*'. I request that, in future, the authors please empathize more with the reviewers.

I appreciate the addition of a Methods section; this is simplistic, but *GC* allows for a pragmatic and *ad hoc* data collection methodology. Attention has been paid to my detailed comment. I am afraid, however, that the authors have failed to make progress with respect to my criticism about 'novelty'. In short, it is not clear from the writing that the outputs of a similar workshop, with identical findings, was not published last year in the *Journal of XXXXXX*.

This could be fixable relatively simply through a number of actions - see detail below (i) a few sentences, (ii) a little stylistic tweaking in places including the abstract, and (iii) some detail to your recommendations. But, this remains a major point in terms of the presentation/framing of the work.

In light of how the response was presented, I started re-reading the abstract, including re-reading the initially submitted abstract. At the end, despite various changes e.g. a '*list of concrete recommendations*' I still have no idea about the novelty in this work. To be specific: How many of these recommendations are new, and how many are simply repeats of previous similar workshops? In reality, if none are new, and all are simply repeats of suggestions in previous work (i.e. our problems have not gone away since 2015), this is fine. But, I personally believe it is necessary to give due credit to past 'state-of-the-subject' workshops and similar if they exist, or be clear if they do not.

The following bullets are suggestions, which I intend to be constructive to fix the 'novelty' issue on the assumption it is presentational (i.e. there isn't a recent similar identical paper).

- Is it that there are no previous/recent/relevant attempts to summarize views on this subject? If so, please state this. If there are, please add a couple of sentences to outline what they are, giving references.
- Is the contribution of this paper that it has a small but convenient literature review that 'describes the progress of open weather and climate science in the context of open science developments in general'? (OPEN SCIENCE section). This could have value.
- Is the contribution a snapshot of expert opinion? (TOWARDS OPEN WEATHER AND CLIMATE SCIENCE section). This will have value if it isn't re-inventing the wheel, with a few sentences are no added to demonstrate/assert that this is not the case.
- You could partially for my concerns about a disconnect/lack of awareness of previous views of challenges/issues by adding some kind of categorization to your list of recommendations (e.g. N = new, R = recent - perhaps last 2 years, O = onging/long-term). A simple round-robin e-mail to the co-authors asking them to assign these, then going with the majority, would work.

I apologise if the tone of this review is 'grumpy'. I think it is fair, but it highlights the potential value in making life easy for reviewers (i.e. some of them may not make the effort to get over their initial mood).

I look forward to re-reading this relatively soon,

John

---

## Author Response (AR3)

Dear editor,

Thank you for your assistance during the process.

As requested, we formatted the list of recommendations as a table.

Kind regards,

Martine de Vos

---

## Author Response (AR4)

**All authors' responses  GC-2019-22**

Weather and Climate Science in the Digital Era, Martine G. de Vos et al.

**First revision**

Reviewer Comment 1

General:

*The paper is an interesting read on an important and timely topic. The length is appropriate, and the authors represent a broad spectrum of subjects within Geosciences. However, the paper could improve significantly if the text would be more specific and if more specific examples would be provided for the claims in the text (see list of points below but there are many more places in the text).*

We agree with the reviewer that we could be more specific on how we have collected our observations and how these support the claims in the text. We will add a dedicated 'Methods' section (see below for the suggested text). Besides, throughout the paper we will rephrase text to be more specific on our observations and how these support our story.

" **Methods :** The focus of the conference session was on data and compute intensive approaches that are applied in weather and climate science. The session comprised 10 oral abstract presentations, one keynote talk, and 6 short poster pitches. The 16 participants were either presenters or involved in the organization of the session, and represented domain science, as well as computer and data sciences.

The first part of the session was dedicated to the presentations.The second part was interactive. In three groups of each 5 or 6 persons the participants discussed the "challenges and opportunities regarding open weather and climate science" and noted their findings on a flipchart. The findings of each group were presented and discussed in a following plenary session. Observations and insights from the plenary discussion were documented. The observations in this paper are based on both the insights from the studies presented in the session, and the notes made during the interactive part of the session. The majority of the participants from the session also contributed to this paper. As such this paper represents a shared view of the participants, i.e., a group of experts in weather and climate science, on the digital and open science developments in their field."

*I think that it would make the paper much more credible if the authors would provide a list of action points to improve the situation at the end of the paper. However, I leave this for the authors to decide.*

We thank the reviewer for this great suggestion. At the end of the paper, we will provide a list of action points or conclusions that are described in the different sections of the paper.

*The English should be improved.*
We agree with the reviewer and get a native speaker to edit the manuscript

**Specific comments:**

• *ll.6-9: How is this shown?*
We will rephrase the paragraph to clarify its meaning and add concrete examples that illustrate the importance of shared data and software:
"The majority of studies (roughly 80 %) presented in the conference session depended in some way or another on shared data and software. For example, many studies included open datasets from disparate sources to improve accuracy of forecasts on the local scale, or to extend analyses beyond the domain of weather and climate. Furthermore, shared software is a prerequisite for the studies that presented systems like a model coupling framework or a digital collaboration platform. Although these studies showed that sharing code and data is important, the consensus among the participants was that this is not sufficient to achieve open weather and climate science and that there are important issues to address."

• *ll. 10-14: What is special about the origin, scalability and legal barriers?*
For instance, many data sources come from private industry who may see a competitive advantage to maintaining privacy. But those data may prove useful to the weather community for improving initial conditions of forecast models. Such corundums may be solved by signing nondisclosure agreements and allow weather service to act as trusted agents who use the data for the public good without disclosing their details.

We will include this explanation in the abstract and in the corresponding sections.

• *ll. 10-14: Why does the complexity limit collaboration? Can you give examples?*
We will elaborate the text in the abstract and the corresponding section. Please see the last comment on software platforms for the text suggestion.

• *l. 14: Why is there a need for new roles?*
Data management and programming have become an integral part of current research practice, and these activities require specific digital skills. It is therefore important to acknowledge and define roles, responsibilities and mandates concerning data stewardship and research software engineering.
The aforementioned trusted agents can also be considered a new role

We will include this explanation in the corresponding sections.

*• l. 36: Was this really both short and long wave? If you refer to the 90s, you should also cite the original papers by Chevallier et al.*
We can confirm that neural nets have been used for both short and long wave radiation. We will rephrase the sentence and add the corresponding references.

*• l. 56: Lagging behind whom? Can you give an example?*
The reviewer rightly points out that it is not clear who, or which field we compare to. In fact, open sharing of data, software and vocabularies is only true common practice in a few fields such as astronomy and genomics. Most scientific fields, including weather and climate science, can be considered lagging behind. Furthermore, the actual point was to show that these weather and climate science are mature in terms of applying digital technologies, while the implementation of open science methodologies is less advanced

We will rephrase the corresponding paragraph accordingly.

*• End of section 1: It would be good to give a hint about the structure of the following. The reader does not know what to expect from the rest of the paper.*
We adopt the advice of the reviewer and will clarify the structure of the rest of the paper at the end of the introduction section

*• l. 71-74: Which or at least how many countries? How many Funders and Research institutes? Can you give examples?*
We will try to be more specific and add examples and references to this paragraph.
"Europe and the United States have made efforts to adapt legal frameworks and implement policy initiatives greater openness in scientific research (OECD, 2015; National Science Foundation, 2018). Several countries provide digital infrastructure based on rich metadata for the resources in the research environment, that support their optimal re-use (Mons, 2017). Examples include the European Open Science Cloud in Europe (Directorate-General,2018) , NIH Data Commons projects in the United States, AARnet in Australia (AARNet, 2018) and the African Data Intensive Research Cloud in South Africa (Simmonds,2016). Funders and research institute have announced policies encouraging, mandating, or specifically financing open research practices (Mckiernan et al.,2016; Wilkinson et al, 2016). Examples include the National Science Foundation (NSF) in the United States (National Science Board,2011), CERN in Switzerland (CERN, 2014), the Netherlands Organization for Scientific Research (NWO) (NWO, 2019) and the United Nations Educational, Scientific and Cultural Organization (UNESCO) (Unesco, 2013)."
Unfortunately, we are not able to provide quantitative information.

Mons, B., Neylon, C., Velterop, J., Dumontier, M., Da Silva Santos, L. O. B., & Wilkinson, M. D. (2017). Cloudy, increasingly FAIR; Revisiting the FAIR Data guiding principles for the European Open Science Cloud. Information Services and Use, 37(1), 49–56. https://doi.org/10.3233/ISU-170824

Directorate-General for Research and Innovation. (2018). Prompting an EOSC in practice. Final report and recommendations of the Commission 2nd High Level Expert Group on the European Open Science Cloud (EOSC). https://doi.org/10.2777/112658

AARNet. (2018). ANNUAL REPORT / 2018 DATA CONNECTOR FOR THE FUTURE. Chatswood, Australia.

CERN-OPEN-2014-049. (2014). Open Access Policy for CERN Physics Publication.

R. Simmonds, Taylor, R., Horrell, J., Fanaroff, B., Sithole, H., Rensburg, S. J. van, & Al., E. (2016). The African data intensive research cloud. IST - Africa Week Conference.

National Science Board. (2011). Digital Research Data Sharing and Management.

NWO executive board. (2019). Connecting Science and Society - NWO strategy 2019-2022.

UNESCO Executive board. (2013). Open Access Policy concerning UNESCO publications.

• *l.82: As you elaborate later on, OpenIFS is not Open Source as it has a (free) license.*
The reviewer is right. We will rephrase the sentence.

• *End of section 2: You could also mention Reanalysis data here.*
We adopt the reviewers suggestion and will add the following text:
"Already since the 1990s the international meteorological and climate research communities started sharing data. Examples of data sharing with common file and metadata formats are reanalysis data, starting with NCEP/NCAR reanalysis and ECMWFs ERA reanalysis data products (e.g. Dee et al 2011, Kalnay et al. 1996) and coupled model intercomparison projects (Taylor et al. 2012)."

• *l.108: "clearly enrich their research" Can you give an example how?*
Examples include the various reanalysis datasets published by the ECMWF and NOAA/NCAR that are made freely available to the community and application of the open models, such as WRF.
We will add these examples to the section.

• *l. 114: What is "CF"?*
CF conventions provide guidelines for the use of metadata in the netCDF file. We will rephrase the paragraph and include the meaning and use of CF

• *l. 124: What do you mean by "performance scalability". Software tools that allow to evaluate data at scale on supercomputers? How is data interoperable?*
• *l. 132: Which tools? Can you name them?*

We will rephrase the corresponding sentences to clarify the challenges of producing FAIR weather and climate model data:

"Regarding open and interoperable weather and climate model data, i.e. data and metadata that are formatted according to community standards (CF, CMIP, WMO), we consider performance scalability as the foremost technological challenge. Whereas high-resolution weather and climate data is predominantly produced on large clusters using many compute nodes, subsequent data processing and analysis is often still confined to a single CPU, and hence does not scale easily with, e.g., increased model resolution. Producing FAIR model data via traditional post-processing pipelines is quickly becoming unfeasible for high-resolution climate model data due to the sheer volume and complexity of the model output as noted above."

*• l. 144: Which Journals? Can you name them?*
We will add some examples to the text:
"Data journals, like Geoscience data journal (Royal Meteorological Society), Scientific Data (Springer Nature) and Earth System Data (Copernicus Publications), are a partial remedy, as these provide open access platforms where scientific data can be peer-reviewed and formally published."

*• l. 150: Can you outline some of the examples in more detail?*
We will elaborate the examples and rephrase the paragraph as follows:
"The conference session provided excellent examples of tools and approaches that were developed and made openly available to the research community. For example, approaches to reduce the computational or post processing costs of existing simulation models (Stringer et al., 2018; Behrens et al., 2018; van den Oord et al., 2018, Jansson et al., 2018) and approaches to integrate data sets from different sources (van Haren et al., 2018; Schultz et al., 2018). Several of the studies in the session presented an approach for which open data and software is a prerequisite, for example because these comprise a model coupling framework or a digital collaboration platform (Pelupessy et al., 2018; Ramamurthy, 2018; Hut et al., 2018; Bendoukha, 2018)."

*• . "The studies show that use of machine learning methods has added value because models are built with data beyond standard meteorological data. For example, local conditions related to the natural and built environment that cannot be captured easily in simulation models can be taken into account through trained models." I do not understand this. Can you rephrase?*
This paragraph is about the use of data beyond the standard meteorological datasets. We will rephrase the paragraph to clarify this.

*• l. 177: Can you name examples for hardware and software platforms. And can you define what you mean by "platform" in this context?*
These platforms refer to digital platforms that use cloud technologies to create a virtual research

environment where scientific end-users can store, analyze and share their data. In the conference session several of these platforms were presented. An example of a current platform is the Open geospatial Consortium. We will rephrase the paragraph to clarify this.

• *"data such as that of the environment and citizen science sources." I do not know which data sets you are referring to here.*
This sentence is referring to the data sets described in the section on open data, i.e., social media posts and observations from amateur weather stations. We will rephrase the sentence to make this clear

• *"The increase in accuracy and skill of forecasts at local scales are shown, improved consistency of data products and improved efficiency and skill of simulations, often crossing different disciplines." Again, I do not understand this. Do you mean "show" instead of "are shown,"?*
The reviewer is right, it should have been "show". We will rephrase the sentence accordingly.

• *l. 194: Which issues?*
This term refers to the issues described in the next paragraphs in the same section. The reviewer rightly points out that this should be clear from the text. We will rephrase the text in the section correspondingly.

• *"Technologically, the promise of using modern digital technologies is not always met due to the complexity of software platforms." I do not understand this.*
The cloud appears to be a potential avenue, as it enables individual researchers to gain access to high computing resources, vast amounts of storage and a suite of software tools. In our session, several digital platforms were presented, that use cloud technologies to create a virtual research environment where scientific end-users can store, analyze and share their data. The participants also observed, however, that current platforms, like the Open Geospatial Consortium and JRC Earth Observation Data and Processing Platform, do not seem to increase the extent of scientific collaboration, especially across disciplines. This may be partly due to the fact that these platforms each have implemented their own set of standards for both data formats and interfaces to access these data. Since scientists are required to invest time and effort in working with a specific platform, the heterogeneity poses hurdles to their collaboration with researchers on another platform.

We will rephrase the paragraph to clarify this:

**Minor points:**

• *l. 9: Rephrase: "that here are"*

We will rephrase the sentence

• l. 32: Rephrase "since ensured"
We will rephrase the sentence

• l.45: Rephrase: "use of using"
We will rephrase the sentence

**Reviewer Comment 2**

**Major Points**

*1. There are a number of typographical mistakes, albeit mainly subtle. So please get a native english speaker to proof-read the manuscript. Namely, I have not attempted to pick up all typos.*
We adopt the advice of the reviewer and get a native speaker to edit the manuscript

*2. The methodology (i.e. what was done in the session) needs to be clarified e.g. (i) were specific questions/topics posed for this research exercise [which it was], (ii) elicitation by sticky notes or hands in the air or by the co-authors making notes of what the group said? I think the observational data are (i) L57-558 - a specific session to discuss (by unstated means) the issues (unspecified in detail), and (ii) L20 insights from the work in the rest of the conference (by unstated means). A 'Methods' section needs to be added, which is one place where the questions asked at the session could be stated.*
*3. The 'novelty' (i.e. what is reported here that is not stated elsewhere) is difficult to distinguish, although a Methods section and taking care to phrase the results/discussion in terms of the evidential basis of insights should fix this.*
*4. The Abstract portrays all the thoughts as entirely new, rather than emerging from a context. e.g. L8 'we observed' - we reaffirm? we agree with the informal subject-wide consensus? Please rephrase where appropriate. As an editor of GC, I note that this was submitted as a review article, but it may be better classified as a standard paper.*
The approach followed in the session was similar to a 'focus group' approach where experts in share views and experiences. This paper is not a classical science paper addressing a well posed problem, but synthesizes those experiences from arguably a wide range of specialists. We agree with the reviewer that both the context and type of this research, and the methodology deserve clarification. We adopt the advice of adding a dedicated 'Methods' section (see below for the suggested text). Besides, throughout the paper we will rephrase text to correctly reflect our methodology.

" **Methods :** The focus of the conference session was on data and compute intensive approaches that are applied in weather and climate science. The session comprised 10 oral abstract presentations,

one keynote talk, and 6 short poster pitches. The 16 participants were either presenters or involved in the organization of the session, and represented domain science, as well as computer and data sciences.

The first part of the session was dedicated to the presentations.The second part was interactive. In three groups of each 5 or 6 persons the participants discussed the "challenges and opportunities regarding open weather and climate science" and noted their findings on a flipchart. The findings of each group were presented and discussed in a following plenary session. Observations and insights from the plenary discussion were documented.

The observations in this paper are based on both the insights from the studies presented in the session, and the notes made during the interactive part of the session. The majority of the participants from the session also contributed to this paper. As such this paper represents a shared view of the participants, i.e., a group of experts in weather and climate science, on the digital and open science developments in their field."

**Minor Points**

*Title - The paper's contents are about open access, not digital (see L2&3). Suggest changing title to reflect this.*

We agree with the reviewer that this paper is about open science. In fact, we think we really do include both open science and the digital era. We suggest that we include both terms in the title, i.e., Open Weather and Climate Science in the Digital Era. In the introduction we will point out what we mean by "digital era".

*L6 - 'the studies in the conference session showed' - How exactly?*

We will rephrase the paragraph to clarify its meaning and add concrete examples that illustrate the importance of shared data and software:

"The majority of studies (roughly 80 %) presented in the conference session depended in some way or another on shared data and software. For example, many studies included open datasets from disparate sources to improve accuracy of forecasts on the local scale, or to extend analyses beyond the domain of weather and climate. Furthermore, shared software is a prerequisite for the studies that presented systems like a model coupling framework or a digital collaboration platform. Although these studies showed that sharing code and data is important, the consensus among the participants was that this is not sufficient to achieve open weather and climate science and that there are important issues to address."

*L8 - 'we observed' - how (in)formally was this done?*
*L62 - A brief comment on the limitations/benefits of the approach used to bring together the information for this paper appears necessary in the Methods section.*
*L99 & 103 - Session/sessions? One 'session' with multiple time blocks?*

*L103 - A hint of what was done. Good, but please expand in a Methods section. Using the standard Method/Results/Discussion format might help the clarity of the work. Having everything merged into thematic section currently makes determining what this paper adds difficult, although by clearly stating which evidence comes from where and moving from data to discussion within the existing sections might also work.*

*L104 - 'Discussed'. Please elaborate. e.g. who is 'we'. The co-authors of this paper? How was it determined what are 'common findings' and 'highlights'?*

*L118 - Example of where evidential basis could be clarified. 'we recognized': we as co-authors discussing and concluding, we in the session, and how was this recognized (e.g. large majority in room, or someone mentioned, or did all participants agree to a circulated notes/minutes?).*

We agree with the reviewer that both the context and type of this research, and the methodology deserve clarification. We adopt the advice of adding a dedicated 'Methods' section (see our reply in 'major points' for the suggested text). Besides, throughout the paper we will rephrase text to correctly reflect our methodology.

*L9 - Typo - 'there' not here*

We will rephrase the sentence

*L11 - 'primarily due to'? i.e. either these were refined from a list for some reason, or is this the complete list of possibilities?*

This statement refers to the section where these barriers are described in more detail. For instance, many data sources come from private industry who may see a competitive advantage to maintaining privacy. But those data may prove useful to the weather community for improving initial conditions of forecast models. Such corundums may be solved by signing nondisclosure agreements and allow weather service to act as trusted agents who use the data for the public good without disclosing their details.

We will include this explanation in the abstract and in the corresponding section.

*L19-20 - It is claimed that 'much faster progress' is being made as 'observed from the studies presented in the conference'. This is quite a leap of logic, and is one illustration of how the manuscript could be more tightly argued and/or presented. If this is simply the authors impression, this is fine, but should be clarified by adding 'we believe' or similar. If written as a statement, and evidential basis should be provided in the new data collected. If this is simply a confirmation of what is in the existing literature (i.e. L52-53) then this should be also clarified.*

The reviewer rightly points out that this is the authors' view. We will rephrase the sentence accordingly.

*L21 - Typo - .. computationally intensive ...*

We will rephrase the sentence

*L22 - Introduction. A wide range of topics and issues are introduced here. They are placed in historical context, which is good. But, the treatment of these becomes quite vague when the actual session is mentioned (L58- 59)*

The introduction discusses the role and use of technology in weather and climate science in history as well as the 'digital era '. We will clarify this as mentioned in the reply on the first comment.

We will move the description of the session to the new Methods section.

*L39-48 - This paragraph is currently un-referenced. Please add these.*

We adopt the reviewers suggestion and will add the following references to the paragraph:

Bauer, P., Thorpe, A., & Brunet, G. (2015). The quiet revolution of numerical weather prediction. Nature, 525(7567), 47–55. https://doi.org/10.1038/nature14956

Huntingford, C., Jeffers, E. S., Bonsall, M. B., Christensen, H. M., Lees, T., & Yang, H. (2019). Machine learning and artificial intelligence to aid climate change research and preparedness. Environmental Research Letters, 14(12), 124007. https://doi.org/10.1088/1748-9326/ab4e55

Schneider, T., Lan, S., Stuart, A., & Teixeira, J. (2017). Earth System Modeling 2.0: A Blueprint for Models That Learn From Observations and Targeted High-Resolution Simulations. Geophysical Research Letters, 44(24), 12,396-12,417. https://doi.org/10.1002/2017GL076101

Reichstein, M., Camps-Valls, G., Stevens, B., Jung, M., Denzler, J., Carvalhais, N., & Prabhat. (2019). Deep learning and process understanding for data-driven Earth system science. Nature, 566(7743), 195–204. https://doi.org/10.1038/s41586-019-0912-1

Ruti, P., Tarasova, O., Keller, J., Carmichael, G., Hov, Ø., Jones, S., … Yamaji, M. (2019). Advancing Research for Seamless Earth System Prediction. Bulletin of the American Meteorological Society, (August 2019), 23–35. https://doi.org/10.1175/bams-d-17-0302.1

L40 - 'exascale' - I don't know this word. Please add a reference or two so that non-specialists can inform themselves.

The term 'exascale' computing refers to $10^{18}$ operations per second, a factor of 1000 beyond current machines.

We will explained the term in the text and add a reference to the sentence:
Reed, D. A., & Dongarra, J. (2015). Exascale computing and big data. Communications of the ACM, 58(7), 56–68. https://doi.org/10.1145/2699414

*L62 - Open science. This appears to be a literature review, unrelated to the session mentioned. Was the session simply used as a brainstorming exercise to get the information together for such literature reviews? If so, again this is fine, but include a Methods section to state this, even if it's only a paragraph long. When the paper is revised, I would expect to distinguish whether the information is (i) in the literature, and being brought together here (ii) views of people in the room etc ...... And, this will allow the contribution of this paper to be clarified/determined. If this*

*section is a review, say 'review' not 'explore', but my Methods points still stand w.r.t later sections.*

The reviewer rightly points out that this section contains a literature review on open science. We will clarify this both in this section, i.e., rephrase the 'explore' sentence, and at the end of the introduction section, where we explain the structure of the paper. We will also add a dedicated 'Methods' section (see our reply in 'major points' for the suggested text).

*L106 - Please try to be specific. Does 'many' mean 5, 50% or something different? It should be possible to give numbers for papers in your session, or you might randomly sample the conference in a desk-based exercise.*

We agree with the reviewer and throughout the paper we will rephrase text to be more specific on our method of data collection.

*General - Is there scope for a table of key points, or graphic to present the most important findings? I am a bit ambivalent about saying this as us readers shouldn't be lazy, but this could usefully highlight the key detailed points. Example of how this could be done - each co-author gets 3 votes, and size of coloured blob relates to number of votes in the graphic.*

We thank the reviewer for this great suggestion. At the end of the paper, we will provide a list of action points or conclusions that are described in the different sections of the paper.

**Short Comment 1**

*P2, L 45. There is a line that talks about the "third development". The construction of this paragraph could be slightly modified to explicitly present the three developments, for a better flow.*

We adopt the suggestion of the reviewer and will modify the construction of the paragraph

*P3, L 63. Section 2, Consider eliminating too many "and" conjunctions.*

We adopt the suggestion of the reviewer and will check the text for unnecessary "and" conjunctions

*P4, L 94-96. Examples or relevant references cited will improve the effectiveness of this statement.*

In fact, open sharing of data, software and vocabularies is only true common practice in a few fields such as astronomy and genomics. Most scientific fields, including weather and climate science, can be considered lagging behind. We will add a few references to support this.

*P4, L 106 onwards. Some parts in 3.1 Open Data seem to fall under 3.2 Open software. But, this could also mean they are very coupled. No changes necessarily needed here.*

*P5, L 118. While interpreting , "Making data and software findable..", software may include tools that lead to the data. I think some level of paraphrasing may be required in this paragraph to make the message from the paper more evident, about making all the components adhere to FAIR goal as a whole.*

The reviewer is right, data and software in are connected and both should adhere to the FAIR principles. We will modify the text of this paragraph (and if necessary other parts of the paper) to clarify this message.

P5, L 126. *This paragraph does provide good insights. But, the final message is not translated well enough as to how this affects open data/science.*
P5, L 131. *Just a note- Removing the need for post-processing by incorporating as many steps as possible within the model itself can make the model computationally even more expensive. Thus, when there is a use-case to share model source code, one may still find it challenging, though open. Though there is one helpful cloud computation reference cited, I would have expected to see more bits about cloud computing in this paper, in this particular section.*

We agree with the reviewer that the impact on open data/science can be stated more clearly. We included a more elaborate description that producing FAIR model data is necessary, but can not be achieved through traditional post-processing pipelines.

Furthermore, we agree with the reviewer that cloud computing technologies, like xarray, Dask, and Apache SPARK, could be useful, since data processing and analysis pipelines usually do not require communication between parallel jobs. One of the key aspects, however, is the capability of the developer, usually a meteorologist or climate scientist, to adopt a new programming paradigm that allows the parallel execution of the workflow on cloud infrastructure. Here research software engineers may play a key role by, e.g., building useful tooling on top of existing low-level platforms like Apache Spark or Dask.

We will rephrase the paragraph accordingly.

*P6, L 161. Punctuation. Add comma after conference.*
We will rephrase the sentence

*P6, L 178 The message/action item here seems to have not translated well here. It does sound contradictory, but the essence of the message might be lost, regarding the technical challenges and reduced scope for multi-discipline collaboration. Please paraphrase this to improve the paragraph.*
We will rephrase the paragraph to clarify the message:

"The use of software as presented above, motivated by open science principles, requires a suitable digital infrastructure. The cloud appears to be a potential avenue, as it enables individual researchers to gain access to high computing resources,

vast amounts of storage and a suite of software tools. In our session, several digital platforms were presented, that use cloud technologies to create a virtual research environment where scientific end-users can store, analyze and share their data. The participants also observed, however, that current platforms, like the Open Geospatial Consortium and JRC Earth Observation Data and Processing Platform, do not seem to increase the extent of scientific collaboration, especially across disciplines. This may be partly due to the fact that these platforms each have implemented their own set of standards for both data formats and interfaces to access these data. Since scientists are required to invest time and effort in working with a specific platform, the heterogeneity poses hurdles to their collaboration with researchers on another platform."

P7, L 194 Punctuation. Replace "here" with "there.
*We will rephrase the sentence*

*P7, L 216 This statement is well put in terms of sharing knowledge. I hope this can be reflected more in the paper.*
We thank the reviewer for this comment. Throughout the paper we will rephrase text to be more specific on our observations and how these support our story. At the end of the paper, we will compile a list of action points or conclusions, i.e., to improve the current situation, that are described in the different sections of the paper.

**Second revision**

**Reviewer 1**

*-I would recommend to shorten the second half of the abstract.*
We agree with the reviewer that a concise abstract is more clear and convenient. However, as we also want to be specific and concrete in our information, which in our case includes providing examples, we have chosen not to shorten the second half of abstract.

*- "weighting of those multiple models" should be rephrased*
We have rephrased the sentence (line 46 )

*- "10ˆ18"*
We have rephrased the sentence (line 53)

*- "three orders of magnitude greater than the speed of current machines" This is incorrect.*
We have rephrased the sentence to : Exascale (i.e., 1018 operations per second) is the next proxy in the long trajectory of exponential performance increases that has continued for more than half a century (Reed,2015) (lines 53-55)

*- "In the section Towards Open Weather and Climate Science" Maybe use italic to highlight that this includes the actual title.*
We have adopted the suggestion (line 154)

*- "In the Methods section" Is there actually a methods section? This section seems to be present in the diff document but missing in the main document.?*
The reviewer is right, we have forgotten to include this new section in the main document. We have included the Methods section in this second revision of the manuscript (lines 82-96)

*- "depending as it does on its intended use" should be reformulated*
We have rephrased the sentence (line 282)

*- "this is true for hardware and software-hardware interaction as well" I do not understand what this means in this context.*
We have rephrased the sentence to : this is true for hardware and the software run by these hardware as well (line 285)

Reviewer 2:
*Firstly, I would like to comment that the authors' response is not particularly kind to reviewers in our attempt to determine if the requested changes have been made. Whilst areas of change have been highlighted, none of the responses are linked to the changes (e.g. line numbers in new document), and my major comments 2-4 with a single comment that is essentially 'Read the new Methods section, and we've made a selection of changes throughout'. I request that, in future, the authors please empathize more with the reviewers.*
We apologize for not being clear to the reviewer how we responded to your valuable comments. We include a more extensive response and references to line numbers now.
I appreciate the addition of a Methods section; this is simplistic, but GC allows for a pragmatic and ad hoc data collection methodology. Attention has been paid to my detailed comment. I am afraid, however, that the authors have failed to make progress with respect to my criticism about 'novelty'. In short, it is not clear from the writing that the outputs of a similar workshop, with identical findings, was not published last year in the Journal of XXXXXX. Our manuscript reflects the current discourse on research software, infrastructure and open science in weather and climate research and the opportunities for sharing and combining data, software and infrastructure. This is an ongoing debate in the community of which aspects are discussed in isolation. Here we report on these discussions as part of eScience developments. Elements have been discussed in literature, e.g. in Ruti et al (2020) on strategic programming level, in Righi et al (2020) on a generic software tool for Earth system model data diagnostics, the open software platform PANGEO (https://pangeo.io/), and community simulation model as the regional model WRF and CESM (Skamarock,2019; Hurrell, 2013). Additionally, these aspects are discussed in Climate Informatics workshops (http://climateinformatics.org/ ), workshops held as part of the European Network on Earth System Modelling (ENES, https://portal.enes.org/ ), workshops of operational centres as the European Centre for Medium-Range Weather Forecasting (e.g. the bi annual High Performance Computing workshop) to name a few. Our

approach in this paper goes further in a) discussion of open science aspects which is developing recently in our field (e.g. FAIR principles on data and research software) and b) the direct relation with compute infrastructures. These aspects make the approach novel.

*This could be fixable relatively simply through a number of actions - see detail below (i) a few sentences, (ii) a little stylistic tweaking in places including the abstract, and (iii) some detail to your recommendations. But, this remains a major point in terms of the presentation/framing of the work. In light of how the response was presented, I started re-reading the abstract, including re-reading the initially submitted abstract. At the end, despite various changes e.g. a 'list of concrete recommendations' I still have no idea about the novelty in this work. To be specific: How many of these recommendations are new, and how many are simply repeats of previous similar workshops? In reality, if none are new, and all are simply repeats of suggestions in previous work (i.e. our problems have not gone away since 2015), this is fine. But, I personally believe it is necessary to give due credit to past 'state-of-the-subject' workshops and similar if they exist, or be clear if they do not.*

See response above, we believe two aspects are novel: 1) the emphasis on open science in weather and climate research, which hardly received attention so far and 2) the consideration of the integration of new developments in data, software and hardware and the challenges and opportunities that come about.

*Indeed, this type of paper, unlike a scientific paper where a hypothesis is developed and tested using data, this paper reflects more the expert opinions of the authors. The following bullets are suggestions, which I intend to be constructive to fix the 'novelty' issue on the assumption it is presentational (i.e. there isn't a recent similar identical paper).*

- *Is it that there are no previous/recent/relevant attempts to summarize views on this subject? If so, please state this. If there are, please add a couple of sentences to outline what they are, giving references.*

We have added references and descriptions to related studies and discussions to the OPEN SCIENCE section (lines 122-129 )

- *Is the contribution of this paper that it has a small but convenient literature review that 'describes the progress of open weather and climate science in the context of open science developments in general'? (OPEN SCIENCE section). This could have value.*
- *Is the contribution a snapshot of expert opinion? (TOWARDS OPEN WEATHER AND CLIMATE SCIENCE section). This will have value if it isn't re-inventing the wheel, with a few sentences are no added to demonstrate/assert that this is not the case.*

We have formulated the contribution and novelty of the paper in paragraphs in both the abstract (lines 22-25) and the discussion (lines 245-250)

- *You could partially for my concerns about a disconnect/lack of awareness of previous views of challenges/issues by adding some kind of categorization to your list of recommendations (e.g. N =new, R = recent - perhaps last 2 years, O =*

> *onging/long-term). A simple round-robin e-mail to the coauthors asking them to assign these, then going with the majority, would work.*

We have added a reflection on novelty per recommendation and add a reference when it is ongoing work.

*I apologise if the tone of this review is 'grumpy'. I think it is fair, but it highlights the potential value in making life easy for reviewers (i.e. some of them may not make the effort to get over their initial mood).*

The review is fair and it is critical for the quality of the peer review system that reviewers are critical.

**Most relevant changes in manuscript**

For detailed information on all changes made in the manuscript, please see the point by point response to the reviewers' comments

- Modified title
- More emphasis on the added value of this work; in the abstract
- Explanation of the paper structure; in the introduction section
- Inclusion of a dedicated method section
- Additional references and concrete examples to place our work in context of international initiatives and similar studies in the domain; in the 'open science' section
- Additional concrete examples and quantitative observations to be more explicit about our methodology; in the section 'towards open weather and climate science'
- Additional explanations on several topics (i.e., machine learning, performance scalability, agile,software platforms); in the section 'towards open weather and climate science'
- Reflect on our methodology; in the 'discussion' section
- Inclusion of a list of action points/ key observations; at the end of the paper

[revised manuscript text omitted]